# Unstable TTTTA/TTTCA expansions in *MARCH6* are associated with Familial Adult Myoclonic Epilepsy type 3

Rahel T. Florian et al.[#]

Familial Adult Myoclonic Epilepsy (FAME) is a genetically heterogeneous disorder characterized by cortical tremor and seizures. Intronic TTTTA/TTTCA repeat expansions in *SAMD12* (FAME1) are the main cause of FAME in Asia. Using genome sequencing and repeat-primed PCR, we identify another site of this repeat expansion, in *MARCH6* (FAME3) in four European families. Analysis of single DNA molecules with nanopore sequencing and molecular combing show that expansions range from 3.3 to 14 kb on average. However, we observe considerable variability in expansion length and structure, supporting the existence of multiple expansion configurations in blood cells and fibroblasts of the same individual. Moreover, the largest expansions are associated with micro-rearrangements occurring near the expansion in 20% of cells. This study provides further evidence that FAME is caused by intronic TTTTA/TTTCA expansions in distinct genes and reveals that expansions exhibit an unexpectedly high somatic instability that can ultimately result in genomic rearrangements.

*email: christel.depienne@uni-due.de  [#]A full list of authors and their affiliations appears at the end of the paper.

FAME is an autosomal dominant, very slowly progressive condition characterized by cortical tremor affecting mainly the hands, frequently associated with generalized myoclonic and sometimes tonic-clonic seizures, and, more rarely, focal seizures[1–3]. This condition was first described in Japan as benign adult familial myoclonic epilepsy (BAFME), and subsequently also referred to as familial cortical myoclonic tremor with epilepsy (FCMTE) or autosomal dominant cortical myoclonus and epilepsy (ADCME). Several different chromosome loci, identified through linkage, at 2p11-2q11, 3q26-q28, 5p15, and 8q24, have been reported[4–7] but the genetic variants underlying the disorder have remained elusive for 20 years despite extensive sequencing of genes contained in these intervals.

Recently, intronic expansions composed of mixed TTTTA/TTTCA repeats in *SAMD12* on chromosome (chr) 8q24 have been identified as the main cause of FAME1 (BAFME1) in the Japanese and Chinese populations[8–11]. *SAMD12* pentanucleotide repeat expansions are associated with a specific haplotype originating from a founder effect in Asia[8,10]. Interestingly, two Japanese families without *SAMD12* expansion had similar TTTTA/TTTCA repeat expansions in *RAPGEF2* (chr4) and *TNRC6A* (chr16)[8].

We previously investigated a large French family with FAME3 (previously referred as FCMTE3, OMIM 613608) linked to a 9.31 Mb region on chr 5p15.31-p15.1[6,12] (Family 1; Fig. 1a). Sequencing of all exons in the linked interval by next generation sequencing had excluded the existence of pathogenic coding variants. Parallel research in a large Dutch FAME pedigree (Family 3; Fig. 1c) linked to the same region on chr5p had revealed a missense variant (NM_001332.3:c.3130G>A, p. Glu1044Lys) in *CTNND2*, which segregated in all affected family members but one, who was considered a possible phenocopy[13,14].

In the present study, we present evidence that FAME3 results from repeat expansions similar to those described in *SAMD12* for FAME1 families, but located at a different site in the first intron of *MARCH6*. These expansions range from 3 to 14 kb on average and show extensive variability in length and structure in blood cells. This instability extends to genomic micro-rearrangements occurring at or near the expansion site in individuals with expansions larger than 10 kb. The mean TTTCA repeat length inversely correlates with the age at seizure onset, providing further evidence that the TTTCA insertion constitutes the pathogenic part of the expansion. We also demonstrate that expansions have no detectable consequence on *MARCH6* expression in blood and skin of affected individuals. The observation of similar repeat expansions in distinct, apparently unrelated genes strongly suggests that these expansions lead to FAME independently of their genome location and impact on the recipient gene.

## Results

**Identification of *MARCH6* expansions in four families.** To identify the pathogenic variant in Family 1, we performed whole genome sequencing and, in parallel, sequenced RNA (PolyA+ and small RNA) extracted from lymphoblastic cells of three affected members and one healthy spouse using short-read Illumina technology (Methods). Combined analysis of genome and RNA-seq data, including detection of structural variants and splicing defects, failed to detect any possible pathogenic variants shared by affected family members or significant alteration of genes in the linked interval (Supplementary Data 1). We then used ExpansionHunter[15] to search for TTTTA/TTTCA repeat expansions within the linked region. This analysis revealed reads with TTTCA repeats mapping to a region composed of 12 TTTTA repeats in the human reference assembly (GRCh37/hg19) located in intron 1 of *MARCH6* (chr5:10,356,460–10,356,519;

Fig. 2a), which was one of the two possible expansion sites predicted by Ishiura and colleagues[8]. TTTCA repeats at this locus were observed in all three affected members of Family 1 but absent from the healthy spouse and individuals from another family (Family 5, Supplementary Fig. 2) linked to the FAME2 locus on chr2[16] (Fig. 2b). Similar results were obtained with exSTRa[17] and STRetch[18] while TTTCA repeats were identified in genomes of both families using TRhist[19] (Supplementary Fig. 1).

Visualization of the mapped reads suggested the following expansion structure: 5′-(TTTTA)$_{exp}$(TTTCA)$_{exp}$-3′. To confirm this result, we set up 5′- and 3′-repeat-primed PCR (RP-PCR) assays using, respectively, reverse (AAAAT) and forward (TTTCA) primers directly binding within the expansion (Fig. 2a). These assays confirmed the existence of 5′-TTTTA and 3′-TTTCA expanded motifs in all 16 affected individuals tested as well as in one unaffected individual (Fig. 1a, Fig. 2c, Supplementary Fig. 3).

We then used the RP-PCR assays to screen the Dutch Family 3 and eleven additional FAME families of European origin (Fig. 1b–d, Supplementary Fig. 2, Supplementary Data 2) for the *MARCH6* expansion. This analysis revealed expansions in Family 3 and two additional families (Fig. 1b–d, Supplementary Fig. 3). The expansion co-segregated with the disorder in all families, including the affected individual 3-IV-9, who did not carry the *CTNND2* p.Glu1044Lys variant (Fig. 1c, Supplementary Fig. 3). This finding led to the reclassification of the *CTNND2* variant as likely benign, despite its impact on neuronal morphology in vitro[13], and to consider the *MARCH6* expansion as the cause of FAME in this family.

**Variability of TTTTA repeat number in control individuals.** Analysis of the region where the expansion occurs in 83 European control individuals from two different cohorts showed that it corresponds to a polymorphic microsatellite (short tandem repeat), with the number of TTTTA repeats typically ranging from 9 to 20 (Fig. 2a; Supplementary Fig. 4a-b). We never observed larger TTTTA repeats or repeats containing TTTCA motifs in control individuals. A similar number of TTTTA repeats is present in Chimpanzees and Bonobos while this number is reduced in more distant primate species (Supplementary Fig. 4c).

**Haplotype analysis reveals an ancient common ancestor.** We used available SNP data from the French families (Families 1 and 2) to investigate the possibility of a common haplotype underlying the expansion. The core haplotype from these two families is located at chr5(hg19):10301295–10492095, and is only 190.8 kb (0.35 cM) in size (Supplementary Fig. 5). It encompasses the entire *MARCH6* gene, as well as two other genes (*ROPN1L* and 5′ of *CMBL*). We calculated that 253.1 generations (confidence interval (CI): 76.1–953.6) separate the two families at this locus. Assuming a 20-year generation span, a common ancestor with this haplotype would have lived ~5060 years ago (CI: 1520–19080).

**Characterization of expansion length and structure.** We next sought to characterize the length and structure of *MARCH6* expansions. Since short-read sequencing data do not permit accurate assessment of repeat number exceeding the corresponding read length, we used long-read Oxford Nanopore Technology to sequence the genome of six individuals from Families 1 and 2. Low-coverage sequencing (6–10×) allowed retrieving one to four reads displaying the expansion per individual (Supplementary Fig. 6). Detected expansions typically spanned 4–6.5 kb and comprised between 791 and 1035 repeats

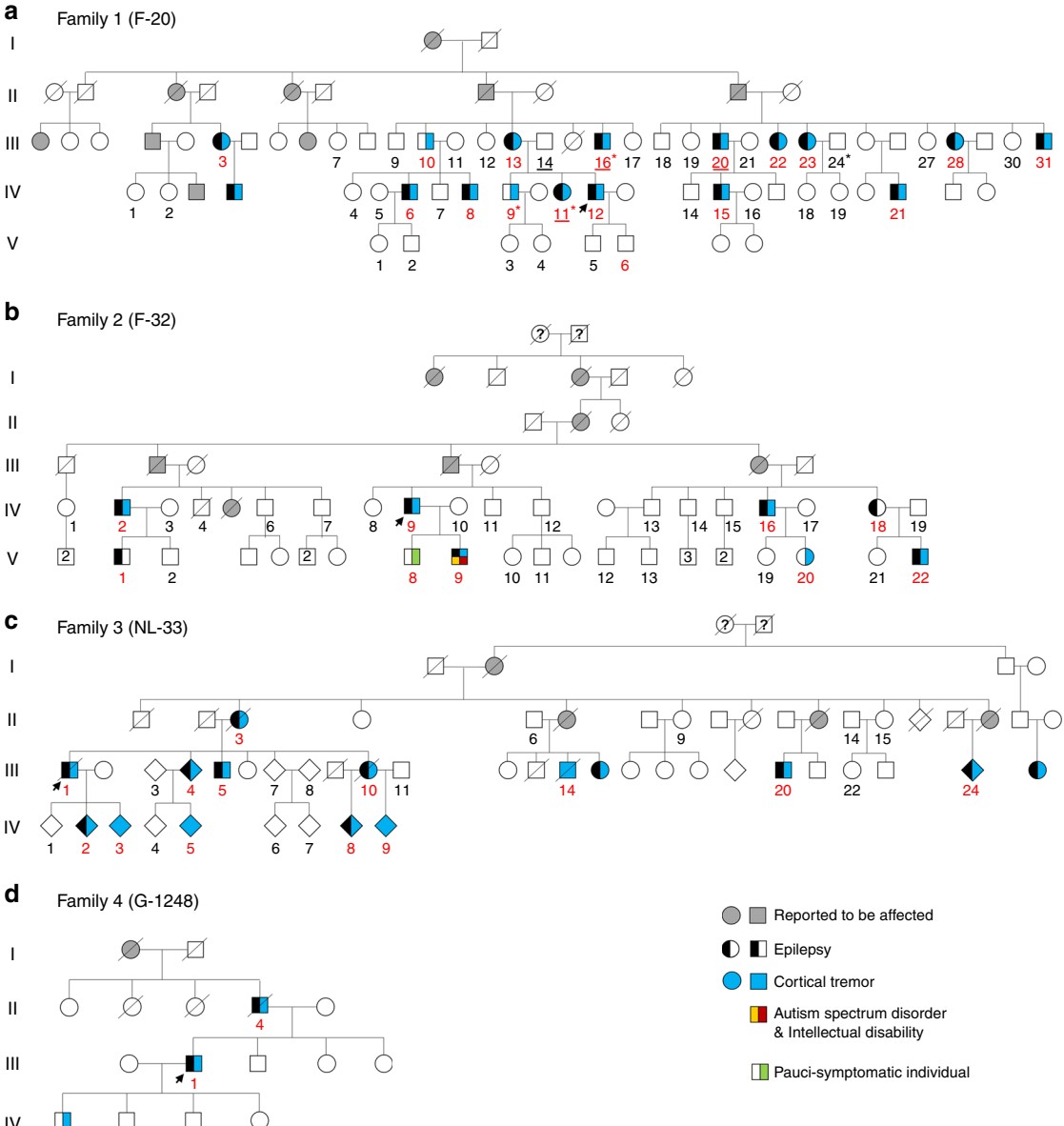

**Fig. 1** Pedigrees of families with *MARCH6* expansions. Pedigrees of Families 1 (**a**, French), 2 (**b**, French), 3 (**c**, Dutch), and 4 (**d**, German). Individuals with ID numbers in red are carriers of the expansions. Individuals with ID numbers underlined have been included in whole-genome sequencing analyses. Individuals with stars have been included in RNA-seq analyses. Black half-filled symbols represent individuals with seizures; Blue symbols indicate individuals with cortical or myoclonic tremor. Individuals with both cortical tremor and epilepsy appear with one half each. A re-examined carrier individual presenting with minor signs of tremor (pauci-symptomatic individual) is indicated with a green half square. One male individual of Family 2 had autism spectrum disorder (yellow corner) and intellectual disability (red corner). Arrows indicate probands. ID numbering in Families 1 and 3 is identical to that previously described[6,14]

in total (Fig. 3a–c, Table 1). However, we observed a substantial variability in reads covering the expansion in the same individual (Fig. 3c, d, Supplementary Data 3, a-i). Four reads incompletely covering the expansion were sequenced in individual 2-IV-9, two of them spanning a variable TTTCA stretch that was alone up to 5 kb (Supplementary Data 3, j–m).

To confirm that the observed variability possibly reflects somatic mosaicism and not an artifact introduced by the sequencing procedure, we used molecular combing (Fiber FISH) to analyze very long, single-stretched DNA fibers in an unbiased fashion in blood cells from nine members of Families 1 and 2 and one healthy control. We stained the TTTCA repeats (in red) and the regions flanking the expansions (in blue and green) by in situ

hybridization (Fig. 4a), and measured the length of every signal for all alleles present on at least one coverslip per individual (i.e., ~100 alleles per coverslip, Supplementary Figs. 7, 8, Supplementary Table 1). This method confirmed the extensive variability in expansion length and structure existing in blood cells of each affected individual (Fig. 4b, c). We recurrently observed staining patterns compatible with different expansion configurations (Fig. 4c, d). The existence of multiple expansion configurations was further supported by positive results using the same RP-PCR assay with either TTTTA or TTTCA-priming oligos in several individuals (Supplementary Fig. 9).

Calculation of the size using molecular combing data showed that expansions range on average from 3.34 to 14.07 kb (Table 1,

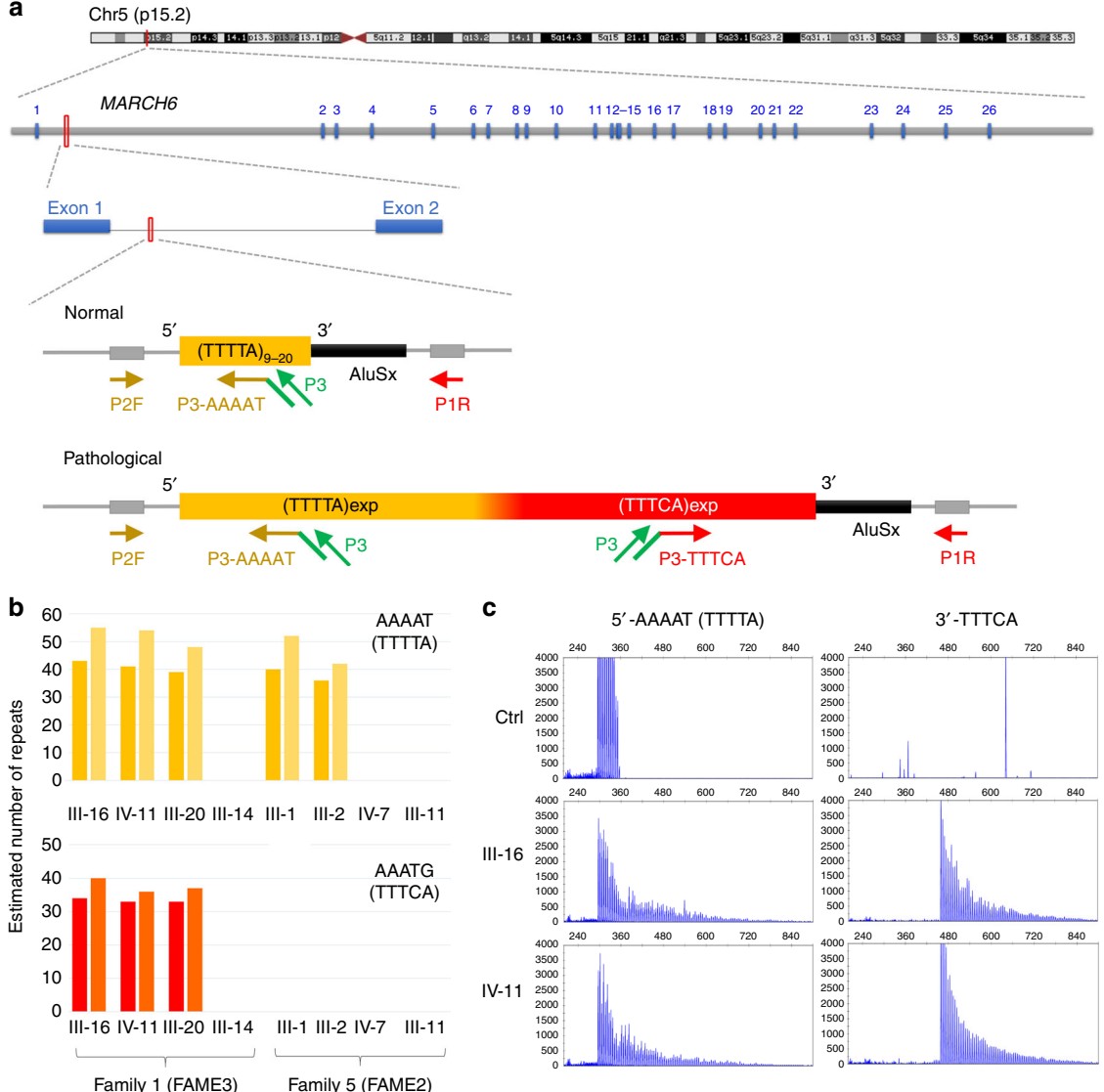

**Fig. 2** Identification of TTTTA/TTTCA expansions in *MARCH6*. **a** Schematic representation of the region where the expansion occurs in intron 1 of *MARCH6* on chromosome 5p15.2. Blue boxes (1–26) represent *MARCH6* exons. The yellow rectangle indicates the TTTTA repeats while the red rectangle represents the TTTCA repeats. Yellow (5′-AAAAT assay) and red (3′-TTTCA assay) arrows indicate primers used for the repeat-primed PCR assays while green arrows schematize the universal primer used in the assay (P3). **b** Number of TTTTA (actual repeated motif searched for: AAAAT, panel above, yellow) and TTTCA (actual repeated motif searched for: AAATG, panel below, red) repeats identified by ExpansionHunter from Illumina short-read genome data of three affected individuals (1-III-16, 1-IV-11, 1-III-20) and one healthy spouse (1-III-14) of Family 1 and another FAME family linked to FAME2 on chr2 (Family 5). Dark and light bars indicate allele 1 and allele 2, respectively. **c** Results of 5′-AAAAT (left panels) and 3′-TTTCA (right panels) RP-PCR assays in a control individual (healthy blood donor) and two affected individuals (1-III-16 and 1-IV-11) of Family 1

Fig. 5a, Supplementary Table 1). The analysis was extended to fibroblasts of the same individuals from Family 1, with similar results (Fig. 5b, Supplementary Table 1), suggesting that the expansions had comparable characteristics in blood and skin.

**Micro-rearrangements are associated with large expansions.** The index case of Family 2 (2-IV-9) and his son (2-V-9) exhibited several DNA molecules harboring complex micro-rearrangements at the expanded site (Fig. 4e, Supplementary Fig. 10), representing up to 10% of alleles present on the coverslip (i.e., in up to 20% of cells, Fig. 4f). Similar micro-rearrangements were observed at a lower frequency in individuals 1-IV-6, 1-IV-8, 2-IV-18, and 2-V-22 (Fig. 4f). One of the nanopore reads covering the expansion in individual 2-IV-9 (read 2-IV-9_2, Supplementary Data 3, k)

spanned the 3′ flanking region and TTTCA part of the expansion on chromosome 5p15.2 fused to a region on chromosome Xp22.3 encoding the uncharacterized LOC107985675 ncRNA gene, suggesting that this read corresponds to a micro-rearrangement involving another chromosome.

The two individuals with frequent micro-rearrangements strikingly had the largest expansions: the father (2-IV-9) had expanded alleles ranging from 1.7 to 36.8 kb, with a mean expansion length of 14.1 kb, while his son had expansions comprised between 5.4 and 30.1 kb, with an average size of 13.3 kb (Fig. 5a, Table 1, Supplementary Table 1). Micro-rearrangements thus likely result from somatic instability, and the frequency of these events appears positively correlated with the expansion size.

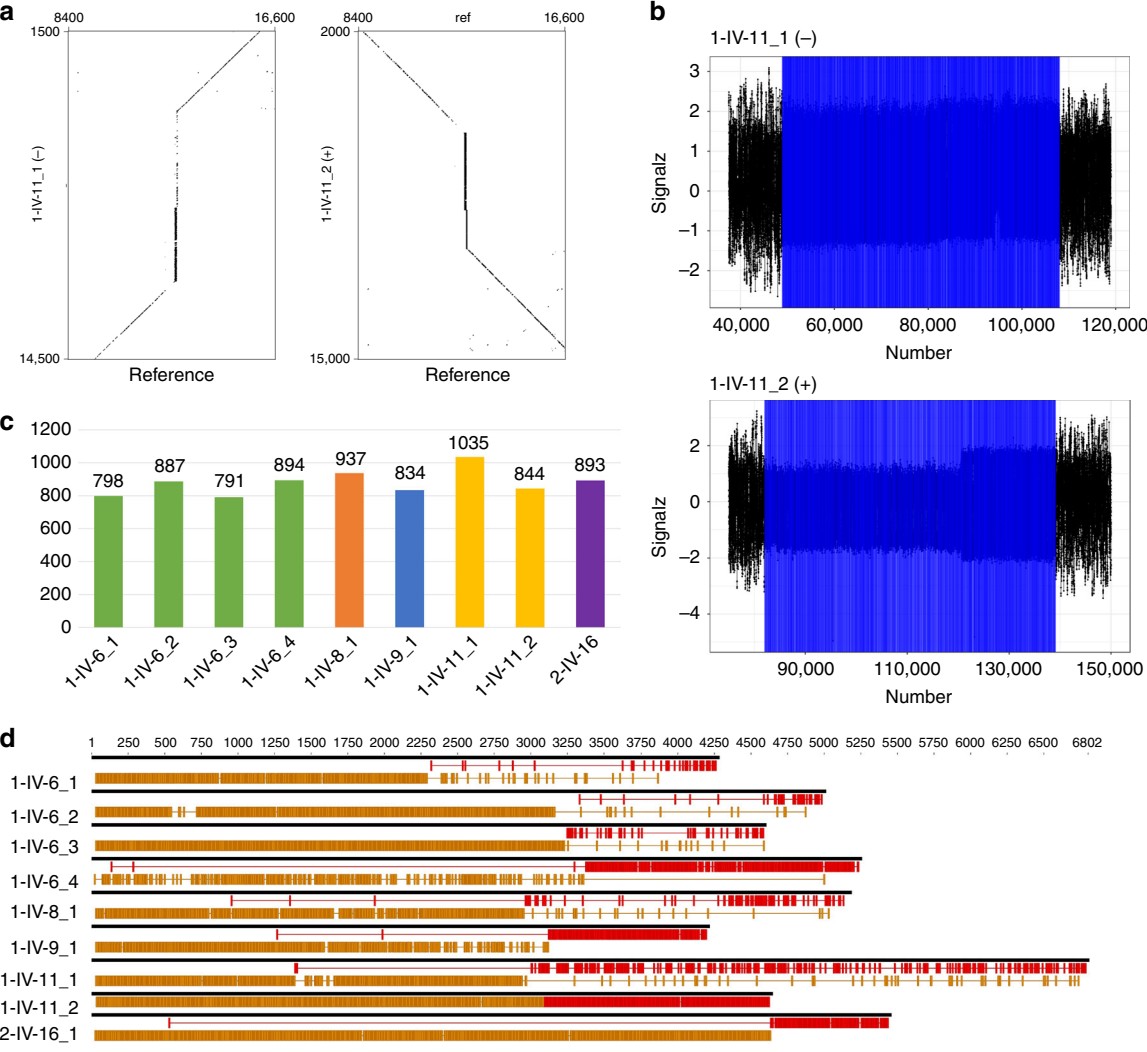

**Fig. 3** Characterization of *MARCH6* expansions by Nanopore sequencing. **a** Dot plots comparing two nanopore reads from individual 1-IV-11 displaying the expansion (*Y*-axis, scale: 13 kb) with the corresponding hg19 reference region (*X*-axis, scale: 8.1 kb). The expansions appear as vertical lines. Read 1-IV-11_1 is on the negative strand while read 1-IV-11_2 is on the positive strand. **b** Analysis of the same raw nanopore reads using NanoSatellite. The signals corresponding to the expanded repeats appear in blue. **c** Number of total repeats inferred by NanoSatellite for each extracted read covering the expansion. Data are displayed for the five individuals for whom reads covering the whole expansion have been detected. Four reads covering parts of the expansion and flanking regions were obtained for individual 2-IV-9 but are not included in this graph. Dot plots and raw nanopore reads covering completely the expansion appear in Supplementary Fig. 6 and all sequences are available in Supplementary Data 3. **d** Schematic representation of the sequence of the same nanopore reads showing exact TTTTA motifs in yellow and exact TTTCA motifs in red. Gaps between exact repeats possibly correspond to interruptions or sequencing (base calling) errors

**Phenotypic variability and genotype–phenotype correlations.** We used data from blood cells to explore further the relationship between the repeat number of each motif and the age at onset of epilepsy and tremor. We observed an inverse correlation between the age at seizure onset and the length of the expansion, mainly driven by the size of the TTTCA repeats (Fig. 5c). On the contrary, no significant correlation was observed between the age at tremor onset and any parts of the expansion (Fig. 5d).

Accordingly, the two individuals with the largest expansions (2-IV-9, 2-V-9) were amongst the most severely affected individuals. Both started to have generalized seizures at 17–18 years of age. Individual 2-IV-9 had a moderate, asymmetric myoclonic tremor affecting the upper limbs (the right side being more affected than the left side) when last examined at age 60 years despite treatment with sodium valproate (VPA) and clobazam (CLB), and he also showed non-specific gait difficulties.

Individual 2-V-9 (28 years old) had autism spectrum disorder (ASD) and intellectual disability (ID) in addition to FAME and lived in an institution for disabled persons. Analysis of trio exome had failed to reveal any other pathogenic variant in this individual and it remained unclear whether his ASD-ID phenotype was related to the FAME phenotype.

Conversely, three individuals harboring expansions were reported asymptomatic at the time of blood sampling (1-IV-8, 1-V-6, 2-V-8) but only two were available for re-examination. Five years after sampling, at age 30 years, individual 2-V-8 (son of 2-IV-9) had discreet signs of tremor and never had seizures (Supplementary Fig. 11). This individual was not included in molecular combing analyses and we could not determine the size of his expansion. Eleven years after the first sampling, at age 53 years, individual 1-IV-8 reported walking difficulties possibly due to myoclonic tremor affecting lower limbs and worsened by

**Table 1 Summarized clinical features and expansion characteristics for 10 affected individuals**

| Family | 1 | 1 | 1 | 1 | 2 | 2 | 2 | 2 | 2 | 2 |
|---|---|---|---|---|---|---|---|---|---|---|
| Patient ID | IV-6 | IV-8 | IV-9 | IV-11 | IV-9 | IV-16 | IV-18 | V-9 | V-20 | V-22 |
| Sex | M | M | M | F | M | M | F | M | F | M |
| Age at last examination | 58 | 53 | 61 | 60 | 60 | 71 | 67 | 28 | 39 | 44 |
| *Clinical features* | | | | | | | | | | |
| Age at tremor onset | 30 | 52 | 25 | 30 | 40 | 14 | – | NA | 14 | 28 |
| Age at seizure onset | 25 | 46 | – | 32 | 18 | 30 | 30 | 17 | – | 30 |
| Symptom at onset | Sz | 1 Sz | CT | CT | Sz | CT | Sz | Sz | CT | Sz/CT |
| *Nanopore sequencing (blood)* | | | | | | | | | | |
| No. of P alleles | 4 | 1 | 1 | 2 | 4 | 1 | ND | ND | ND | ND |
| Mean expansion size | 4.73 | 5.13 | 4.16 | 5.67 | NA | 5.40 | | | | |
| Mean 5′-TTTTA size | 2.95 | 2.93 | 3.08 | 3.00 | NA | 4.60 | | | | |
| Mean TTTCA size | 1.78 | 2.21 | 1.08 | 2.67 | >5 | 0.80 | | | | |
| *Molecular combing (blood)* | | | | | | | | | | |
| No. of P alleles | 71 | 58 | 25 | 29 | 219 | ND | 50 | 54 | 30 | 38 |
| Mean expansion size | 4.62 | 5.06 | 3.34 | 4.92 | 14.07 | | 5.72 | 13.33 | 6.16 | 7.55 |
| Mean 5′-TTTTA size | 0.88 | 2.33 | 0.57 | 1.66 | 2.37 | | 3.47 | 1.99 | 2.81 | 3.04 |
| Mean TTTCA size | 2.82 | 2.28 | 2.10 | 2.86 | 10.37 | | 1.99 | 10.04 | 2.93 | 3.60 |
| Mean 3′-TTTTA size | 0.92 | 0.46 | 0.67 | 0.40 | 1.32 | | 0.27 | 1.31 | 0.41 | 0.90 |
| *Molecular combing (Fibros)* | | | | | | | | | | |
| No. of P alleles | 10 | 11 | 41 | 13 | ND | ND | ND | ND | ND | ND |
| Mean expansion size | 6.93 | 4.65 | 3.82 | 4.12 | | | | | | |
| Mean 5′-TTTTA size | 0.13 | 1.04 | 0.02 | 0.50 | | | | | | |
| Mean TTTCA size | 4.56 | 2.63 | 2.73 | 2.04 | | | | | | |
| Mean 3′-TTTTA size | 2.24 | 0.97 | 1.08 | 1.59 | | | | | | |

Ages are expressed in years and expansion size are in kb
*CT* cortical tremor, *F* female, *M* male, *P* pathogenic, *NA* not available, *ND* not done, *Sz* seizure(s)

intermittent photic stimulation. He had a single initially focal, evolving to bilateral convulsive seizure at age 46 years. He was treated with a low dose of VPA and had no further seizures. Neurological examination revealed a mild myoclonic tremor asymmetrically affecting the left upper limb and the right lower limb, without any other symptom. This individual had an expansion of ~5 kb (5.13 kb calculated with Oxford nanopore and 5.06 kb calculated by molecular combing) and a TTTCA length (2.21 kb) comparable to those of three close relatives (1-IV-6, 1-IV-9, 1-IV-11), who were all earlier and more severely affected. This suggests that, although the TTTCA size of the expansion is on average inversely correlated to the age at seizure onset, the symptoms at onset, progression and severity of the disorder are possibly influenced by other factors than the expansion itself or that the expansion sizes observed in peripheral tissues do not accurately predict those existing in the brain.

**Expansions do not alter *MARCH6* expression in blood and skin.** Finally, we investigated whether expansions affect the expression of *MARCH6* in blood cells and fibroblasts of affected family members. *MARCH6* is a ubiquitously expressed gene encoding an E3 ubiquitin ligase that mediates the degradation of misfolded or damaged proteins in the endoplasmic reticulum[20,21]. At least three isoforms, resulting from alternative splicing of exons 2–4 have been detected in several tissues, but the full-length (NM_005885.3) is the predominant isoform (Fig. 6a, GTEx database).

RNA-seq had previously failed to detect any difference in *MARCH6* expression in lymphoblasts of patients and control individuals (Supplementary Fig. 12a). Furthermore, no reads corresponding to *MARCH6* mRNA or small RNA with TTTTA or TTTCA could be detected in these cells (Supplementary Fig. 12c-d). To confirm these findings, we used real-time qRT-PCR with four different primer pairs either overlapping exons 7–8 or exons 14–15, or specifically amplifying intron 1 before or

after the expansion. Both exonic assays showed no difference in *MARCH6* RNA levels in total RNA isolated from blood cells ($n = 12$) or fibroblasts ($n = 4$) of expansion carriers compared with non-carrier individuals ($n = 10$ and 4, respectively; Fig. 6b, c). In agreement with these results, the MARCH6 protein was present at similar levels in fibroblasts of expansion carriers and control individuals (Supplementary Fig. 12b). The intronic assays showed that RNA molecules containing intron 1 are much less abundant compared with the spliced transcripts (~30 times lower), suggesting that these assays detect the transient precursor mRNA. No difference in the level of intron 1-containing RNAs was detected in blood cells of expansion carrier versus non-carrier individuals (Fig. 6d) while a slight decrease was detected in fibroblasts (Fig. 6e), thus ruling out a massive accumulation of abnormally spliced *MARCH6* mRNA carrying the expansion in these cells.

## Discussion

In this study, we provide evidence that FAME3 is due to TTTTA/TTTCA repeat expansions in intron 1 of *MARCH6*. We show that these expansions are somatically unstable in blood cells and fibroblasts, leading to a wide range of expansion sizes and configurations, and that this instability extends to genomic rearrangements in individuals with very large (>10 kb) expansions. Although these genomic rearrangements likely have a deleterious impact on the corresponding cells, it remains unclear, however, whether they directly or indirectly contribute to the pathophysiology of FAME. We confirmed that there is a significant correlation between the size of the expansion and the age at epilepsy onset, as previously reported for *SAMD12* expansions[8], and we now demonstrate that this correlation is mainly due to the size of the TTTCA repeats. This finding provides additional evidence that TTTCA repeats are the pathogenic part of the expansion. One distinctive feature of individuals with *MARCH6* expansions compared with patients with expansions in other FAME genes is

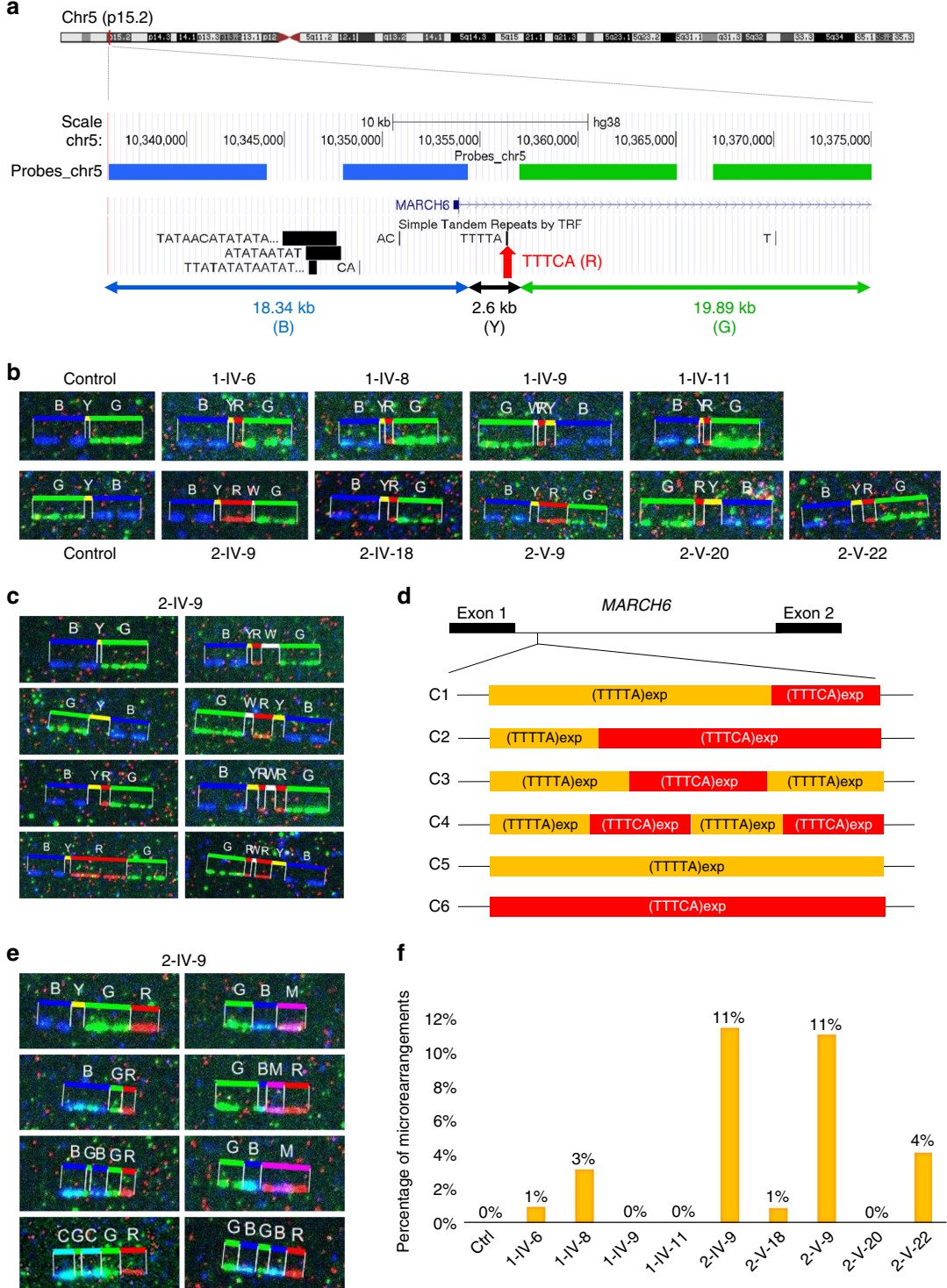

**Fig. 4** Somatic mosaicism of *MARCH6* expansions detected by molecular combing. **a** Schematic representation of the molecular code used to stain regions adjacent to *MARCH6* expansions. Probes directed against specific 5′ (labeled in blue, B) or 3′ (labeled in green, G) flanking regions have been hybridized to single-stretched DNA fibers extracted from blood cells; TTTCA repeats are stained in red (R). **b** Representative images seen in a control individual (two panels on the left) and in nine expansion carrier individuals for whom molecular combing was performed. Y refers to the unstained part between the blue and red signals; unstained parts detected between the red and green signals or in-between two red signals are referred to as W. **c** Selected images observed at the expanded site in the proband of Family 2 (2-IV-9), showing extreme variability of the expansion length and structure in his blood. **d** Schematic representation of the different expansion configurations (C1–C6) observed using molecular combing. **e)** Selected micro-rearrangements observed at the expanded site in individual 2-IV-9. M (magenta) and C (Cyan) correspond to the overlay of red and blue or green and blue probes, respectively, indicating an overlap of probes that should normally be separated. All images corresponding to micro-rearrangements observed in individuals 2-IV-9 and 2-V-9 are shown in Supplementary Fig. 10. **f** Percentage of micro-rearrangements observed in the ten individuals analyzed by molecular combing. Individuals with the largest expansions (2-IV-9 and 2-V-9) exhibit a higher percentage of rearranged alleles than individuals with smaller expansions

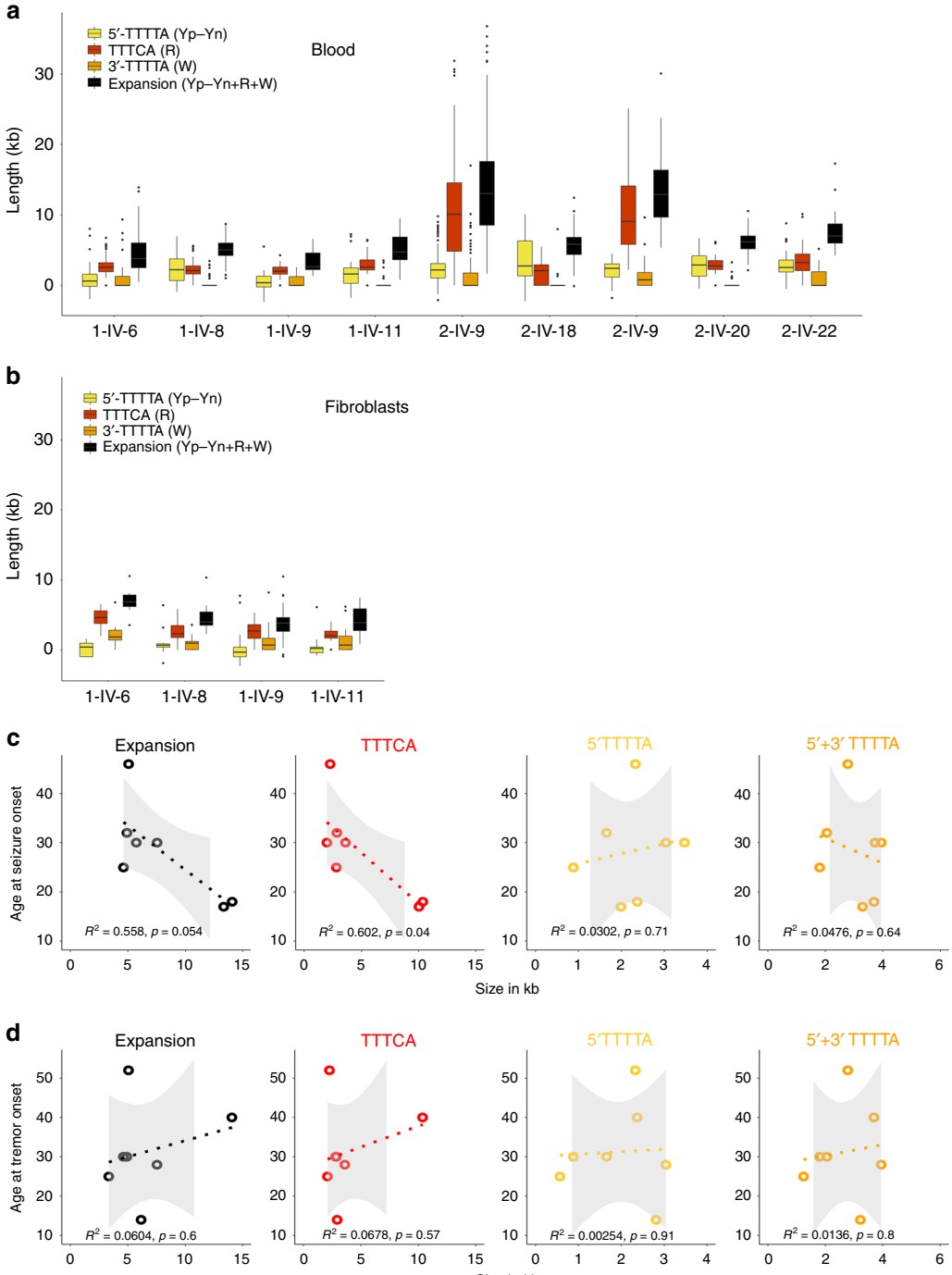

**Fig. 5** Distribution of expansion lengths and genotype–phenotype correlations. **a** Box plots showing the distribution of the size of the overall expansion (in black), as well as the 5′-TTTTA (yellow, Yp–Yn; see Methods for details) and TTTCA (red, R) parts in blood from the nine carrier individuals. Some alleles showed an unstained part between the red and the green signals, which is referred to as 3′-TTTTA (W, in orange). Box plots elements are defined as follows: center line: median; box limits: upper and lower quartiles; whiskers: 1.5× interquartile range; points: outliers. **b** Box plots showing the distribution of the size of the overall expansion (in black) and each parts: 5′-TTTTA appears in yellow (Yp–Yn; see Methods for details), TTTCA in red (R) in fibroblasts from four affected individuals of Family 1. **c** Correlations between the age at seizure onset and the mean size (in kb) of the overall expansion (left), the TTTCA (middle left), the 5′-TTTTA (middle right), or the overall (5′+3′) TTTTA repeats region (right). Individuals with larger TTTCA repeat region have an earlier age at seizure onset. On the contrary, neither the size of 5′-TTTTA or 5′+3′-TTTTA repeats correlate with the age at epilepsy onset. Individuals included in the graph are 1-IV-6, 1-IV-8, 1-IV-11, 2-IV-9, 2-IV-18, 2-V-9, and 2-V-22. Individuals without epilepsy also have the smallest TTTCA stretches although they are not included (see Table 1). $R^2$ is the square value of the Pearson coefficient; 95% confidence intervals appear in gray; corresponding $R$ values and 95% confidence intervals are summarized in Supplementary Table 2. **d** Correlations between the age at tremor onset and the mean size (in kb) of the expansion and each part, showing no correlation with any of them. Individuals included in the graph are 1-IV-6, 1-IV-8, 1-IV-9, 1-IV-11, 2-IV-9, 2-V-20, and 2-V-22

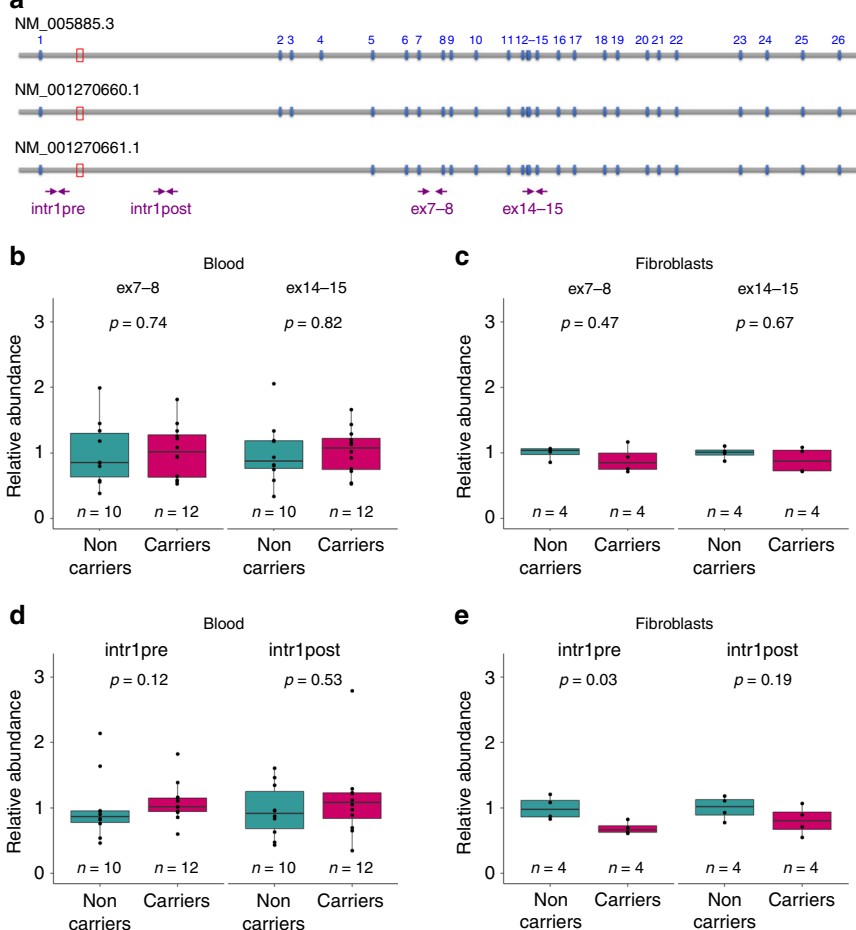

**Fig. 6** Expansions do not affect *MARCH6* expression in blood or skin. **a** Schematic representation of the *MARCH6* transcript isoforms. The site of the expansion is indicated by the red box. Arrows indicate primer pairs used to quantify *MARCH6* gene expression. **b** Results of real-time RT-PCR in blood from expansion carrier (n = 12) versus healthy (n = 10) individuals with primers specific of exons 7–8 (left) and exons 14–15 (right). **c** Results of real-time RT-PCR in fibroblasts from expansion carrier (n = 4) versus unrelated control (n = 4) individuals with primers specific of exons 7–8 (left) and exons 14–15 (right). **d** Results of real-time RT-PCR in blood from expansion carrier (n = 12) versus healthy (n = 10) individuals with primers located in intron 1 before (left) or after (right) the expansion. **e** Results of real-time RT-PCR in fibroblasts from expansion carrier (n = 4) versus unrelated control (n = 4) individuals with primers located in intron 1 before (left) or after (right) the expansion. Box plots elements are defined as follows: center line: median; box limits: upper and lower quartiles; whiskers: 1.5× interquartile range; all values are displayed as points; outliers are shown as disconnected points. Statistical comparisons were done using a Wilcoxon–Mann–Whitney rank-sum test (two-sided)

that seizures precede the onset of tremor in many family members[6,12], but it is unknown whether FAME3 patients have larger TTTCA repeats than other FAME subtypes.

In a companion study, we describe the identification of identical expansions in the first intron of *STARD7* as the cause of FAME2[22]. *STARD7* encodes a ubiquitous protein involved in lipid transport and metabolism[23]. The association of expansions in apparently unrelated genes with similar phenotypes strongly suggests that the pathological mechanism is independent from the gene itself or its function and are more likely related to the type of expansion.

All FAME-related expansions are located within gene introns, suggesting that transcription is a key step in the pathogenic process. Indeed, a common feature of the five genes identified so far harboring FAME expansions (*MARCH6, RAPGEF2, SAMD12, STARD7,* and *TNRC6A*) is their relatively high expression in the human brain, although some genes are more specifically expressed in the central nervous system while expression of others is more ubiquitous. Interestingly, similar intronic TTTTA/TTTCA expansions in *DAB1* have previously been associated with

spinocerebellar ataxia 37 (SCA37)[24]. The difference in phenotype might be attributed to the highly specific expression of *DAB1* in the cerebellum, but several genes where FAME expansion occurs (e.g., *MARCH6, STARD7* and *TNRC6A*) are also highly expressed in the cerebellum. This suggests that the expression profile of the gene where the expansion occurs is important but does not suffice by itself to determine the clinical presentation.

Although *MARCH6* is ubiquitously expressed, our results indicate that the expansion does not alter mRNA and protein levels in blood cells and fibroblasts of carrier individuals compared with those of non-carrier controls. We could not detect either an increase in intron 1 retention that would be expected if RNA molecules containing repeats would accumulate or RNA foci would form. Furthermore, no reads with TTTTA or TTTCA repeats corresponding to *MARCH6* transcripts were detected in lymphoblasts of patients. These results contrast with previous observations made in post-mortem brains of patients with *SAMD12* expansions where reads filled with TTTTA/TTTCA repeats were detected[8], and RNA foci associated with abortive transcription following *SAMD12* expansions were observed[8]. This

discrepancy could be the reflect of processes occurring only in neuronal cells, although this question clearly needs to be further addressed, ideally in additional human brain samples or appropriate cellular organoid models.

Finally, we showed that FAME3 expansions, like FAME1 and FAME2 expansions[8,22], are associated with a common haplotype. However, we calculated that this haplotype comes from an ancestor that would have lived several thousand years ago. It remains unclear whether the expansion was already present on the haplotype at that time, as this would assume that it has not been, or only poorly, counter-selected for more than 200 generations. Another possibility is that repeats would have expanded independently from the same predisposing haplotype more recently. Further investigations are needed to fully understand the precise mechanisms by which similar pentanucleotide repeat expansions in different genes occur and lead to FAME.

## Methods

**FAME families and patients**. Families 1 and 2 are seemingly unrelated French families. Family 1 comprises 24 affected members (16 affected individuals sampled), including 21 with cortical tremor and epilepsy and 3 with cortical tremor only. The clinical features of this family have previously been reported[12]. Genome-wide linkage in this family allowed to identify the FAME3/FCMTE3 locus on chr5p15[6]. Family 2 comprises 14 affected family members (9 affected individuals sampled). Clinical data of this family as well as Families 5, 7–11 were briefly reported[25]. Clinical data of Family 3, originating from the Netherlands, were independently reported[13,14]. Genome-wide linkage in this family was consistent with linkage to chr5p15 and a *CTNND2* missense variant (incompletely) segregating with the disorder was identified[13]. Family 4 is a previously unreported German kindred comprising four affected members, two of whom were available for genetic analyses. The index case, age 54 at the time of the study, suffered from tremor since age 37 and epileptic seizures since age 41. Epileptic seizures occurred in his father and grandmother. First seizures of his father occurred at age 40–45 and he also had a rest tremor. In addition, the eldest son of the index case, at the time of the last follow-up, 32 years old, was affected by tremor since several months, but no seizures were reported. Updated clinical features of Families 1–4 are summarized in Supplementary Data 2. Informed consents were obtained from all participants before sampling. Genetic studies were initially approved by local ethics committee in France (Hôpital Pitié-Salpêtrière, Paris) and Germany (Marburg and Frankfurt hospitals). In the Netherlands, the medical ethical committees of the Academic Medical Centre (Amsterdam UMC) and the Leiden University Medical Centre (CME P117/98) approved the study. The overall study was further approved by the ethics committee of the University Hospital Essen (Germany) in April 2018.

**Whole genome sequencing**. One microgram of genomic DNA extracted from blood samples of eight individuals, three affected individuals (1-III-16, 1-IV-11, and 1-III-20) and one healthy spouse (1-III-14) of Family 1[6], as well as three affected members (5-III-1, 5-III-2, and 5-IV-7) and one healthy spouse (5-III-11) from the FAME2 (chr2)-linked Family 5[16], was used to prepare libraries for whole genome sequencing, using the Illumina TruSeq DNA PCR-Free Library Preparation Kit, according to the manufacturer's instructions. After normalization and quality control, qualified libraries were sequenced on a HiSeqX5 platform from Illumina (Illumina Inc., CA, USA), as paired-end 150-bp reads. One lane of HiSeqX5 flow cell was used for each sample, in order to reach an average sequencing depth of 30X per sample. Sequence quality parameters were assessed throughout the sequencing run and standard bioinformatics analysis of sequencing data was based on the Illumina pipeline to generate FASTQ files for each sample. Reads were then aligned on the human (GRCh37) and decoy (Heng Li's hs37d5 genome for 1000 genomes project [ftp://ftp.ncbi.nlm.nih.gov/1000genomes/ftp/technical/reference/phase2_reference_assembly_sequence/hs37d5.fa.gz] was performed using BWA software (mem+default option [https://github.com/lh3/bwa]). Duplicate sequences were removed from bam files using Sambamba tools [http://lomereiter.github.io/sambamba/docs/sambamba-view.html]. An additional realignment step was performed on the bam file using GATK (RealignerTargetCreator/IndelRealigner). Coverage analyses were generated using an in-house pipeline based on metrics generated by Bedtools programs [http://code.google.com/p/bedtools/]. Variant calling was performed using four tools: UnifiedGenotyper and HaplotypeCaller from GATK, Platypus [http://www.well.ox.ac.uk/platypus], and Samtools [http:///www.htslib.org/]. Results generated by these four programs were assembled in a VCF file. Annotation of the VCF file was carried out and annotated using snpEff and snpSift [http://snpeff.sourceforge.net and http://snpeff.sourceforge.net/SnpSift.html] based on data available in the Ensembl [http://www.ensembl.org/index.html] and dbNSFP [https://sites.google.com/site/jpopgen/dbNSFP] databases.

In an additional step, we screened genome data for tandem repeat expansions using four different bioinformatics pipelines: ExpansionHunter v2.5.5[15], STRetch[18], TRhist[19], and exSTRa[17,26]. We also used ExpansionHunter v2.5.5 to assess the number of TTTTA repeats present in available genome data of 53 control individuals (Supplementary Fig. 4b). By convention, the repeat motif searched for is indicated as the first non-redundant repeat motif using alphabetical order of nucleotides (e.g., AAAAT for TTTTA and AAATG for TTTCA).[27]

**RNA sequencing**. In parallel to genome sequencing, RNA-seq and small RNA-seq was performed for 10 individuals: three affected individuals (1-III-16, 1-IV-9, and 1-IV-11) and one healthy spouse (1-III-24) from Family 1, three affected individuals (5-III-2, 5-III-3, and 5-IV-4) from Family 5 and three healthy controls (Ctrl1–3). Total RNA including small RNAs were extracted from immortalized lymphoblastic cells from each individual using the AllPrepR DNA/RNA/miRNA kit (Qiagen). RNA-Seq libraries were generated from 600 ng of total RNA using TruSeq Stranded mRNA Sample Preparation Kit (Part Number RS-122-2101, Illumina). Polyadenylated RNA (mRNA) were isolated on oligo-d(T) magnetic beads and fragmented at 94 °C for 2 min with divalent cations. First-strand cDNA were synthetized from fragmented RNA fragments using a combination of reverse transcriptase and random primers. Second-strand cDNA synthesis was performed using DNA Polymerase I and RNase H and replacing dTTP with dUTP. A single 'A' base and adapters were successively added on the double-stranded cDNA products, before purification and amplification using the following conditions: 30 s at 98 °C; [10 s at 98 °C, 30 s at 60 °C, 30 s at 72 °C] × 12 cycles; 5 min at 72 °C. Oligos in excess were removed using AMPure XP beads (Beckman Coulter). Capillary electrophoresis was used to check the quality and estimate the quantity of final cDNA libraries. Libraries were sequenced on Illumina HiSeq 4000 sequencer as paired-end 100-base reads following Illumina's instructions at the GenomEast platform (IGBMC, Illkirch, France). Image analysis and base calling were performed using RTA 2.7.3 and bcl2fastq 2.17.1.14. Reads were mapped onto the hg38 assembly of *Homo sapiens* genome using TopHat[28,29] version 2.0.14 and Bowtie version 2-2.1.0[30]. Reads mapping to rRNA and Spikes were discarded. Gene expression was quantified using HTSeq-0.6.1 [http://www-huber.embl.de/users/anders/HTSeq/doc/overview.html] with gene annotations from Ensembl release 88. Statistical analysis was performed using R and DESeq2 1.10.1 Bioconductor library[31]. The differential expression analysis in DESeq2 uses a generalized linear model (GLM) where counts are modeled using a negative binomial distribution. Counts were normalized from the estimated size factors using the median ratio method and a Wald test was used to test the significance of GLM coefficients. Alternative splicing analysis was performed by JunctionSeq[32] version 1.6.0 and rMATS[33] version 3.2.5. Variants were identified using GATK[34] version 3.4-46. At first, duplicate reads were marked using Picard Tools version 1.122. Reads with N operators in the CIGAR strings were split into component reads and trimmed to remove any overhangs into splice junctions. Base quality recalibration and variant discovery process (HaplotyteCaller) were performed. VCF files were filtrated according to the clusters having at least three SNPs in a window of 35 bases between them. Variant annotation was carried out by GATK, SnpEff[35], and SnpSift[36].

Small RNA libraries were generated from 2 µg of total RNA using TruSeq Small RNA Sample Prep Kit (RS-200-0012/0024). The protocol uses RNA molecules that have a 5′-Phosphate and a 3′-hydroxyl group. RNA adapters were ligated to the extremity of RNA molecules, in two different steps: the 3′ RNA adapter, modified to specifically target small RNAs including microRNAs, is added before the 5′ RNA adapter. We subsequently performed a reverse transcription followed by amplification with primers annealing to the adapter ends (30 s at 98 °C; [10 s at 98 °C, 30 s at 60 °C, 15 s at 72 °C] × 13 cycles; 10 min at 72 °C) to selectively enrich RNA fragments containing adapter molecules on both ends. The last step was an acrylamide gel purification of 140–145 nt amplified cDNA constructs (corresponding to cDNA inserts+120 nt adapters). Final libraries were checked for quality and quantified using capillary electrophoresis. The libraries were sequenced on Illumina HiSeq 4000 sequencer as single-read 50-base reads following Illumina's instructions. Image analysis and base calling were performed using RTA 2.7.3 and bcl2fastq 2.17.1.14. Adapters were trimmed from total reads with FASTX_Toolkit [http://hannonlab.cshl.edu/fastx_toolkit/]. Non-coding RNA profiling was performed by the ncPRO-seq 1.6.5 analysis pipeline[37] with the annotation from miRBase release 21[38] and Rfam v11 database[36]. The DESeq2 package v1.16.1[31] was used to normalize and to assess miRNA differential expression between patients and controls.

**Fragment analysis and repeat-primed PCR (RP-PCR)**. Specific primers (FAME3-P2F: CCATCAGAGGCAAGCAATGT and FAME3-P1R: GGGAAAAGGGAGGGGTTATAGAGGA) were designed to amplify the tandem repeat region in intron 1 of *MARCH6* using primer3 [http://primer3.ut.ee/]. Amplicons were analyzed by polyacrylamide capillary gel electrophoresis on an ABI 3130xl DNA Analyzer (Applied Biosystems), according to the manufacturer's instructions. Fragment sizing was performed using GeneMarker (Softgenetics). The repeat expansion was amplified by repeat-primed PCR (RP-PCR) with 6-FAM-labeled FAME3-P2F, P3 (TACGCATCCCAGTTTGAGACG), and either P3-AAAAT (TACGCATCCCAGTTTGAGACGAATAAAATAAAATAAAATAAAATAA) or P3-AAATG (TACGCATCCCAGTTTGAGACGAAATGAAATGAAATGAAA

TG) (5′ assays) or 6-FAM-labeled FAME3-P1R, P3 and either P3-TTTCA (TACG CATCCCAGTTTGAGACG-TTCATTTCATTTCATTTCATTTC) or P3-TTTTA (TACGCATCCCAGTTTGAGACG-TTTTATTTTATTTTATTTTATTTTATTTTA ATTTTA) primers (3′ assays). PCR was performed with 100 ng genomic DNA, 0.8 μM primer FAME3-P2F or FAME3-P1R, 0.8 μM primer P3-PU, and 0.08 μM primer P3-TTTCA or P3-AAATG or 0.27 μM primer P3-AAAAT or P3-TTTTA using the HotStarTaq Master Mix (QIAGEN). The PCR program used with the P3-TTTCA or P3-AAATG primers was 95 °C for 15 min, followed by 40 cycles (94 °C for 1 min, 58 °C for 1 min, and 72 °C from for 2 min 30 s) and a final extension step (72 °C for 10 min). The PCR program used with the P3-AAAAT or P3-TTTTA primers was 95 °C for 15 min, followed by 40 cycles (94 °C for 1 min, 54 °C for 1 min, and 60 °C from for 2 min 30 s) and a final extension step (60 °C for 10 min). RP-PCR products were detected on an ABI 3130xl DNA Analyzer and analyzed using GeneMapper® software v5.0 (Thermo Fisher Scientific).

**Sanger sequencing**. Sanger sequencing was used to determine the number of TTTTA repeats present on non-pathogenic alleles in 30 healthy blood donors (Supplementary Fig. 4a) and FAME3 patients. After amplification of the corresponding region using primers FAME3-P2F and FAME3-P1R, forward and reverse sequence reactions were performed with the Big Dye Terminator Cycle Sequencing Ready Reaction Kit (Applied Biosystems) using the same primers. G50-purified sequence products were run on an ABI 3130xl DNA Analyzer (Applied Biosystems) and sequence data were analyzed with Geneious (Biomatters).

**Haplotype analysis**. SNP genotypes were available for individuals of Families 1 and 2[25]. A core haplotype was defined based on sharing amongst affected individuals from both French families. Haplotype dating was performed using a published method[39], and website developed in the Bahlo lab [https://shiny.wehi.edu.au/rafehi.h/mutation-dating/]. This method determines the age of the most recent common ancestor (MRCA) from whom the core haplotype was inherited. This method can also be used for individuals with shared 'extended haplotypes', who are likely to have a MRCA who is more recent than that for the whole group.

**Oxford nanopore sequencing**. DNA extraction was done from 5 mL fresh blood samples using a modified salting-out procedure[40]. Ten micrograms of DNA was sheared in 100 μL of water by pulling it ten times through a HPLC injection needle (blunt metal needles [91029] Hamilton, attached to a 1 mL BD Luer Lock syringe) and cleaned up with 0.4X AMPure XP beads. Approximately 3.5–5 μg of DNA was used for the library preparation with the 1D Ligation Sequencing Kit (SQK-LSK108) and sequenced on the GridION sequencer utilizing R9.4.1 flow cells. Base calling was done using guppy (Oxford Nanopore Technologies), adapters were trimmed by porechop and reads were filtered with NanoFilt[41] for quality >8. Alignment was done by minimap2[42] and bam files were generated using samtools[43]. Structural variants were called using NanoSV[44] and picky[45]. Tandem repeat lengths within the extracted reads were further analyzed by NanoSatellite[46].

**Molecular combing (Fiber FISH)**. Freshly sampled blood cells from nine patients (1-IV-6, 1-IV-8,1-IV-9, and 1-IV-11 from Family 1, 2-IV-,9 2IV-18, 2-V-9, 2-V-20, and 2-V-22 from Family 2) and one control individual (healthy blood donor) were embedded in agarose plugs using the FiberPrep® DNA extraction kit from Genomic Vision (Bagneux, France). DNA was purified by proteinase K and sarkosyl treatment overnight according to manufacturer instructions. Agarose was melted and digested by an overnight beta-agarase treatment. Purified DNA was diluted in MES buffer and combed on CombiCoverslips® using a FiberComb® (both from Genomic Vision), immobilized on the surface by baking at 60 °C for 4 h and stored at −20 °C until further use. FiberProbes® DNA FISH probes were designed and labeled by Genomic Vision. Briefly, overlapping probes corresponding to 5′ and 3′ regions flanking the expansion were amplified by long-range PCR and purified and used as templates for probe labeling by random priming and combined to a probe directed against the TTTCA part of the expansion (biotin-TTTCATTTCATTTCATTTCATTTCATTTCA). Hybridization was carried out overnight in a Hybridizer (Dako) with the labeled probes, detected using fluorophore-coupled antibodies layers (BA-0500 Biotinylated Anti-Streptavidin Antibody, VECTOR Laboratories, dilution 1:25). The entire coverslip was scanned on a FiberVision® automated scanner (Genomic Vision). Image analysis and signals measurement was performed using FiberStudio® software (Genomic Vision). We measured each part of the signals (B: blue, R: red, G: green) for all alleles present on one coverslip (healthy control and individuals 1-IV-6, 1-IV-9, 1-IV-11, 2IV-18, 2-V-9, 2-V-20, 2-V-22) or two coverslips (individuals 1-IV-8 and 2-IV-9). For alleles without red staining, the unstained part between the blue and green signals was referred to as $Y$ on images. For alleles with red staining (pathogenic alleles), Y refers to the unstained part between the blue and red signals; unstained parts were also detected between the red and green signals or in-between two red signals and referred to as $W$. Distribution of non-pathogenic versus pathogenic alleles was assessed by comparing data of the control individual to those of affected patients (Supplementary Fig. 7). Based on the distribution observed in the control, three categories of alleles without red staining were defined: normal (N) alleles ($Y < 5.5$ kb, $n = 113$ in Ctrl), undefined (U) alleles ($5.5 \leq Y < 8.5$, $n = 1$ in Ctrl) and

likely pathogenic (P) alleles ($Y \geq 8.5$, $n = 0$ in Ctrl) (Supplementary Fig. 7a, c). Only definite pathogenic (with red staining) and likely pathogenic (without red staining; $Y \geq 8.5$) were included in further calculations of expansion length (Fig. 5a, b, Supplementary Fig. 8, and Supplementary Table 1). Rearranged and incomplete alleles were annotated and counted for each patient but excluded from further calculations. For each pathogenic allele, the total size of the expansion was calculated using the formula: $Y_p - Y_n + R + W$, where $Y_p$ is the individual measure of the allele and $Y_n$ is the median $Y$ value of the normal alleles from the same individual. Allele distributions, calculations (mean, median, and standard deviation), graphs (box plots), and statistical analyses were performed using in-house R scripts.

**Real-time quantitative reverse transcription PCR (qRT-PCR)**. MARCH6 isoforms and their corresponding expression were assessed using Alamut 2.11 (Interactive Biosoftware) and the GTEx database [https://gtexportal.org/home/]. Blood samples from 12 expansion carriers (1-IV-6, 1-IV-9, 1-IV-11, and all nine expansion carriers from Family 2) and 10 non-carrier individuals were collected in Paxgene® Blood RNA tubes (PreAnalytiX, Qiagen). Total RNA was isolated using the PAXgene® Blood RNA Kit (Qiagen) and RNA integrity was verified on a 2100 Bioanalyzer (Agilent). In parallel, fibroblasts from four expansion carriers (1-IV-6, 1-IV-8, 1-IV-9, 1-IV-11) and four unrelated healthy individuals were cultured in AmnioMAX (Thermo Fisher Scientific), and total RNA was isolated using the RNeasy mini kit (Qiagen). Primers allowing specific amplification of exons 7–8 (Ex7F-GGAGGAAGATGACGCTGGT, Ex8R-AAAGCATTCCAATTCATGTCA TC) and exons 14–15 of MARCH6 (Ex14F-AATTGGAGTATTCCCTCTCATTTG, Ex14/15R-CAGAGTAGCATCAAACATTTCCA), and primers amplifying intron 1 before (intr1preF2-TGAGGAAACTGATGGTTAGTATGATT, intr1preR2-CTC TGACAGACATGAGTCTGAATCT) or after the expansion (intr1postF3-TTGTT GTGAATGGCTGGATG, intr1postR3-AGGTGCGGATCAGTCCTACA) were designed using the Universal ProbeLibrary Assay Design Center (Roche). Efficiency of each primer pair was first verified using serial dilutions of cDNA (for exonic assays spanning introns) or genomic DNA (for intronic assays) of control samples. To eliminate possible contamination of extracted RNA by genomic DNA, 1 μg of total RNA of each sample was treated for 30 min at 37 °C with RQ1 RNase-Free DNase (Promega) before proceeding to reverse transcription. cDNAs were synthesized using the LunaScript RT supermix Kit (New England Biolab). Reverse-transcribed MARCH6 cDNA was quantified using the LightCycler 480 Probes Master Mix from Roche and Universal Probe Library specific probes. PPIA (F-AT GCTGGACCCAACACAAAT, R-TCTTTCACTTTGCCAAACACC) was used as the control gene. Each sample was run in triplicate on a Lightcycler 480 (Roche) with the following thermocycling conditions: 95 °C for 10 min (1 cycle); 95 °C for 15 s and 60 °C for 1 min (45 cycles); and 37 °C for 30 s (1 cycle). Relative abundance was calculated using the formula $2^{\Delta\Delta Ct} = (Ct_{MARCH6} - Ct_{PPIA})_{individual\ tested}/mean\ (Ct_{MARCH6} - Ct_{PPIA})_{control\ individuals}$. For blood samples, samples ($n = 22$) were run on two different plates each including two MARCH6 specific primer pairs and PPIA, as well as the same two individuals (one expansion carrier, one non-carrier) to control for experiment reproducibility between plates. Fibroblasts ($n = 8$) samples were run on the same plate. Value distributions were compared using a Wilcoxon–Mann–Whitney rank-sum test (two-sided).

**Western blotting**. Fibroblast cells ($1 \times 10^6$) from four expansion carriers of Family 1 (1-IV-6, 1-IV-8, 1-IV-9, 1-IV-11) and four healthy individuals were lysed in 100 μL of NP-40-buffer (500 mM NaCl, 20 mM Tris pH 8, 1 mM EDTA pH 8, 0.5% NP-40) supplemented with Halt Protease Inhibitor Cocktail (Thermo Fisher). Proteins were separated on 10% SDS polyacrylamide gels and transferred to nitrocellulose membrane (GE Healthcare). We used the bs-9340R polyclonal antibody (Bioss Antibodies, dilution 1:300) to reveal MARCH6 and an anti-β-Tubulin (#2146, Cell Signaling Technology; dilution 1:1000) as loading control. We used ImageJ to quantify protein expression.

**Reporting summary**. Further information on research design is available in the Nature Research Reporting Summary linked to this article.

## Data availability
Families included in this study have not consented to have their genome data publicly released. The source data underlying Figs. 1, 2b, 3c, 4f, 5a–d and 6b–e, as well as Supplementary Figs. 4a, 7, 8 and 12a–b are provided as a Source Data file. Raw images (molecular combing experiments) and raw nanopore data (corresponding to reads included in this study) are available from the corresponding author, upon request. The MARCH6 expansion has been deposited in ClinVar under the accession SCV000924549. RNA-seq and small RNA-seq data have been deposited in the ArrayExpress database at EMBL-EBI (www.ebi.ac.uk/arrayexpress) under accession numbers E-MATB-8300 and E-MTAB-8301 .

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

## Acknowledgements

We thank the families for their participation in this study, Agnès Rastetter (ICM, Paris, France) for RNA extraction, and Emmanuelle Apartis (Hôpital Saint-Antoine, Paris, France) for electrophysiological assessment of Family 1. DNA extraction and cell culture of lymphoblasts have been performed at the DNA and cell bank of ICM (Paris, France). RNA-seq has been performed on the GenomEast platform of IGBMC, Illkirch, France. WGS has been performed by the Centre National de Recherche en Génomique Humaine (CNRGH) Institut de Biologie François Jacob, Evry, France. We thank Jean-Louis Mandel and Nicolas Charlet-Berguerand (IGBMC, Strasbourg, France), Cécile Cazeneuve (Hôpital Pitié-Salpêtrière, Paris, France), Charles Marcaillou (Integragen, Evry, France) and Isabel Silveira (Porto, Portugal) for valuable discussions. This study has been financially supported by three different grants from the Fondation Maladies rares to C.D. (2009, 2010, 2016), Assistance Publique des Hôpitaux de Paris (APHP), INSERM, the "Investissements d'Avenir" programme ANR-10-IAIHU-06 (IHU-A-ICM), University Duisburg-Essen and University Hospital Essen. M.B. was supported by an Australian National Health and Medical Research Council (NHMRC) Program Grant (GNT1054618) and an NHMRC Senior Research Fellowship (GNT1102971). This work was also supported by the Victorian Government's Operational Infrastructure Support Program and the NHMRC Independent Research Institute Infrastructure Support Scheme (IRIISS). Laura Canafoglia: Member of the European Reference Network on Rare and Complex epilepsies, ERN EpiCARE.

## Author contributions

R.T.F.: acquisition, analysis, and interpretation of clinical data (Family 2), RP-PCR and molecular combing data, and drafting of the paper. F.K., S.G., and I.K.: acquisition, analysis, and interpretation of nanopore data. E.Lei.: Statistical analysis and development of R pipelines. S.Ka.: acquisition, analysis, and interpretation of RP-PCR, qPCR and western blot data. B.K., C.N., and D.B.: acquisition and analysis of linkage, Sanger and NGS data. S.F., L.J., and R.K.: cell culture, and acquisition of RNA-seq or qRT-PCR data. T.Y., D.P., and B.J.: acquisition, analysis, and/or interpretation of RNA sequencing data, and drafting the corresponding methods. J.-F.D.: acquisition of genome sequencing data. J.B., C.S., V.M., A.M., T.B., M.F.B., H.R., and M.B.: bioinformatic analysis, interpretation of genome and/or haplotype data. J.A., N.T., Y.D., and M.D.M.A.: contributions to experimental design and acquisition of molecular combing data. T.K. and L.S.: contributions to experimental design, acquisition, and analysis of qRT-PCR and western blot data. E.M., A.L., and M.V.: acquisition, analysis and interpretation of clinical data (Family 1). S.Kle. and L.T.: acquisition, analysis, and interpretation of clinical data

(Family 2). P.L.: contribution to initial project design, acquisition, analysis and interpretation of clinical data (Families 1, 2, 5, 12). A-F.v.R., L.S.V., M.A.J.T., and A.M.J.M.v. d.M.: acquisition, analysis, and interpretation of clinical data (Families 3, 6, 9). T.K.: RP-PCR screening of Family 4. P.S.R., F.R., and K.M.K.: acquisition, analysis, and interpretation of clinical data (Family 4). G.R. and E.H.: acquisition, analysis and/or interpretation of clinical data (Families 7, 8, 10, 11). C.G.: acquisition, analysis and interpretation of clinical data (Family 13). E.LeG. and B.H.: contributions to project conception and/or experimental design. M.A.C. and J.G.: coordination of FAME consortium, formulation of theory and prediction, acquisition, analysis, and interpretation of data (Family 4). C.D.: formulation of theory and prediction, contributions to experimental conception and design, acquisition, analysis and/or interpretation of data, coordination of the overall study, drafted the article. FAME consortium: sharing of data, ideas, and resources. All authors and contributors critically revised the paper for intellectual content.

## Competing interests

Y.D. and M.D.M.A. were employed by Genomic Vision, a company providing diagnostic kits for the detection of genomic rearrangements, at the time of the study. All other authors declare no competing interests.

## Additional information

Rahel T. Florian[1], Florian Kraft[2], Elsa Leitão[1], Sabine Kaya[1], Stephan Klebe[3], Eloi Magnin[4], Anne-Fleur van Rootselaar[5], Julien Buratti[6], Theresa Kühnel[1], Christopher Schröder[1], Sebastian Giesselmann[2], Nikolai Tschernoster[7], Janine Altmueller[7], Anaide Lamiral[4], Boris Keren[6], Caroline Nava[6,8], Delphine Bouteiller[8], Sylvie Forlani[8], Ludmila Jornea[8], Regina Kubica[1], Tao Ye[9], Damien Plassard[9], Bernard Jost[9], Vincent Meyer[10], Jean-François Deleuze[10], Yannick Delpu[11], Mario D.M. Avarello[11], Lisanne S. Vijfhuizen[12], Gabrielle Rudolf[9,13], Edouard Hirsch[13], Thessa Kroes[14], Philipp S. Reif[15,16], Felix Rosenow[15,16], Christos Ganos[17], Marie Vidailhet[8,18], Lionel Thivard[18], Alexandre Mathieu[19], Thomas Bourgeron[19], Ingo Kurth[2], Haloom Rafehi[20,21,22], Laura Steenpass[1], Bernhard Horsthemke[1], FAME consortium, Eric LeGuern[6,8], Karl Martin Klein[15,16,37], Pierre Labauge[38], Mark F. Bennett[20,21,22], Melanie Bahlo[20,21], Jozef Gecz[14,39], Mark A. Corbett[14], Marina A.J. Tijssen[40], Arn M.J.M. van den Maagdenberg[12,41] & Christel Depienne[1,8,9]

[1]Institute of Human Genetics, University Hospital Essen, University of Duisburg-Essen, Hufelandstraße 55, 45147 Essen, Germany. [2]Institute of Human Genetics, Medical Faculty, RWTH Aachen University, 52062 Aachen, Germany. [3]Department of Neurology, Universitätsklinikum Essen, Universität Duisburg-Essen, Hufelandstraße 55, 45147 Essen, Germany. [4]Department of Neurology, CHU Jean Minjoz, 25000 Besançon, France. [5]Departments of Neurology and Clinical Neurophysiology, Amsterdam UMC, University of Amsterdam, Amsterdam Neuroscience, Meibergdreef 9, 1105 AZAmsterdam, The Netherlands. [6]AP-HP, Hôpital Pitié-Salpêtrière, Département de Génétique, 75013 Paris, France. [7]Cologne Center for Genomics, Center for Molecular Medicine Cologne (CMMC), University of Cologne, Weyertal 115b, 50931 Cologne, Germany. [8]Institut du Cerveau et de la Moelle épinière (ICM), Sorbonne Université, UMR S 1127, Inserm U1127, CNRS UMR 7225, F-75013 Paris, France. [9]IGBMC, CNRS UMR 7104/INSERM U1258/Université de Strasbourg, 1 Rue Laurent Fries, 67400 Illkirch-Graffenstaden, France. [10]Centre National de Recherche en Génomique Humaine (CNRGH), Institut de Biologie François Jacob, CEA, Université Paris-Saclay, F-91057 Evry, France. [11]Genomic Vision, 80 Rue des Meuniers, 92220 Bagneux, France. [12]Department of Human Genetics, Leiden University Medical Center, Albinusdreef 2, 2333 ZA Leiden, The Netherlands. [13]Department of Neurology-centre de référence des epilepsies rares, University Hospital of Strasbourg, 1 Avenue Molière, 67200 Strasbourg, France. [14]School of Biological Sciences, School of Medicine and Robinson Research Institute, The University of Adelaide, Adelaide 5005 SA, Australia. [15]Epilepsy Center Frankfurt Rhine-Main, Department of Neurology, Goethe University and LOEWE Center for Personalized Translational Epilepsy Research (CePTER), 60323 Frankfurt am Main, Germany. [16]Department of Neurology, Epilepsy Center Hessen, Philipps University, 35037 Marburg, Germany. [17]Department of Neurology, Charité University Medicine Berlin, 10117 Berlin, Germany. [18]APHP, Hôpital Pitié-Salpêtrière, Département de Neurologie, 75013 Paris, France. [19]Human Genetics and Cognitive Functions, Pasteur Institute, UMR3571 CNRS, Université de Paris, 75015 Paris, France. [20]Population Health and Immunity Division, The Walter and Eliza Hall Institute of Medical Research, Parkville 3052 VIC, Australia. [21]Department of Medical Biology, University of Melbourne, Melbourne 3010 VIC, Australia. [22]Epilepsy Research Centre, Department of Medicine, University of Melbourne, Austin Health, Heidelberg 3084 VIC, Australia. [37]Departments of Clinical Neurosciences, Medical Genetics and Community Health Sciences, Hotchkiss Brain Institute & Alberta Children's Hospital Research Institute, Cumming School of Medicine, University of Calgary, 2500 University Dr NW, Calgary, AB T2N 1N4, Canada. [38]Department of Neurology, Gui de Chauliac University Hospital, 34295 Montpellier, France. [39]South Australian Health and Medical Research Institute, The University of Adelaide,

Adelaide 5005 SA, Australia. [40]Department of Neurology, University Medical Center Groningen, University of Groningen, 9700 AB Groningen, the Netherlands. [41]Department of Neurology, Leiden University Medical Center, Albinusdreef 2, 2333 ZA Leiden, The Netherlands. A full list of consortium members appears at the end of the paper.

## FAME consortium

Samuel F. Berkovic[23], Francesca Bisulli[24,25], Francesco Brancati[26,27], Laura Canafoglia[28], Giorgio Casari[29], Renzo Guerrini[30], Hiroyuki Ishiura[31], Laura Licchetta[24,25], Davide Mei[30], Tommaso Pippucci[32], Lynette Sadleir[33], Ingrid E. Scheffer[23,34], Pasquale Striano[35], Paolo Tinuper[24,25], Shoji Tsuji[31] & Federico Zara[36]

[23]Epilepsy Research Centre, University of Melbourne, Melbourne, Australia. [24]IRCCS, Istituto delle Scienze Neurologiche di Bologna, Bologna, Italy. [25]Department of Biomedical and Neuromotor Sciences, University of Bologna, Bologna, Italy. [26]Medical Genetics, Department of Life, Health and Environmental Sciences, University of L'Aquila, L'Aquila, Italy. [27]Laboratory of Molecular and Cell Biology, Istituto Dermopatico dell'Immacolata, IDI-IRCCS, Rome, Italy. [28]Neurophysiopathology, Fondazione IRCCS Istituto Neurologico Carlo Besta, Milan, Italy. [29]Vita-Salute San Raffaele University, Milan, and TIGEM - Telethon Institute of Genetics and Medicine, Naples, Italy. [30]Neuroscience and Neurogenetics Department, Meyer Children's Hospital, Florence, Italy. [31]Department of Neurology, The University of Tokyo Hospital, Tokyo, Japan. [32]Medical Genetics Unit, Sant'Orsola-Malpighi University Hospital, Bologna, Italy. [33]Department of Paediatrics and Child Health, University of Otago, Wellington, Wellington, New Zealand. [34]Austin Health, Australia and Royal Children's Hospital, Murdoch Children's Research Institute and Florey Institute, Melbourne, Australia. [35]Pediatric Neurology and Muscular Diseases Unit, Department of Neurosciences, Rehabilitation, Ophthalmology, Genetics, Maternal and Child Health, University of Genoa, IRCCS Istituto G. Gaslini, Genova, Italy. [36]Laboratory of Neurogenetics and Neuroscience, IRCCS Isttiuto G. Gaslini, Genova, Italy

