## [Peer Review File · Nature Communications]

Reviewers' Comments:

Reviewer #1:

Remarks to the Author:

In this study Florian et al, describe a TTTTA/TTTCA pentanucleotide repeat expansion in the intron 1 region of MARCH6 gene in chromosome 5p15 as a cause of Familial Adult Myoclonic Epilepsy in multiple unrelated European families. Overall the study presents convincing evidence supporting TTTTA/TTTCA pentanucleotide repeat as the cause of FAME. This study is particularly important as it further strengthens the evidence of previous findings that describe same pentanucleotide repeat in the intron regions of three unrelated genes as the causative agent of FAME/BAFME. Overall the study is well done and adds significantly to our understanding of nucleotide repeat expansion disorders more broadly and FAME in particular. Additional experiments are recommended to better illustrate their findings and fit the scope of this journal.

Concerns:

1. Multiple studies finding same repeat expansion in different introns leading to same disease strongly indicates TTTTA/TTTCA repeat RNA toxicity in FAME/BAFME. In fact, earlier published studies for the same repeat showed RNA foci/aggregate formation. Although extensive mechanistic studies are beyond the scope of this manuscript, the lack of any data supporting expression of the repeat make these data difficult to interpret. A number of steps could be taken to address this:
 - a. Is there any evidence that the TTTCA repeat-containing intron is transcribed or retained in patient cells at increased rates compared to control cells?
 - c. Does the expansion affect mature mRNA and MARCH6 protein expression? These can be easily determined by RT-PCR and western blotting (using commercially available antibodies).
 - d. Perhaps beyond the scope of this manuscript, Does this TTTCA RNA form nuclear foci in patient cells or tissue samples?
2. Three individuals harboring the repeat expansion reported asymptomatic, two of which later showed mild symptoms. Is there any other factor such as age (25 yrs) or repeat size that can explain this individual (1-V-6)?

Minor points:

3. How many individual samples are used for Oxford Nanopore sequencing? It appears six individuals from 2 families are used, but in the figure legend (Figure 2D) it says five. Also, first individual in the same figure is labeled as 1-III-6_1, which should be 1-IV-6_1.
4. What is the correlation (r^2 value) between mean size of the repeat and age of onset of seizure (Figure 3D)? What is the 95% confidence intervals for this correlation? Which individual samples are used to calculate these correlations? These factors should be delineated in Figure 3 legends.
5. Family 2 proband (2-IV-9) sample was used to test mosaicism by molecular combing (Figure 4A), but it is unclear why 2-IV-9 sample was not included in following RP-PCR test (Figure 4C).
6. Figure 4E needs more detailed description. It seems observed micro-arrangements in individual 2-IV-9 are noted as 'M' (Stained in Pink color?), but there is no description. Also, in the figure legend 4E appears as 4F and 4F appears as 4G.

Reviewer #2:

Remarks to the Author:

Florian et al show that an expansion in the intron of MARCH6 is responsible for the familial adult myoclonic epilepsy linked to chromosome 5 (FAME3). The researchers performed WGS in a family with multiple members affected with FAME3. Concurrently, they performed RNA-Seq and they did not identify any obvious slicing variants; however, they re-analyzed their WGS data with ExpansionHunter. They specifically searched for TTTTA/TTTCA expansions given the knowledge of FAME1 and additional TTTTA/TTTCA on chromosomes 16 and 4 in families with FAME. This analysis showed the expansion in intron 1 of MARCH6 and this segregated with FAME3 in 4 families. The repeat expansion is present in the unaffected population and range from 9-20 repeats but is between 791 and 1035 repeats in 6 cases. They noted that these expansions showed a significant degree of somatic mosaicism and further genomic rearrangement (by combing).

Comments:

The study is convincing, well-presented and results are interpreted appropriately. There is no doubt that this expansion causes FAME3.

I really have only one comment and that is the lack of experimental evidence for how this expansion causes the disorder FAME3.

The authors comment on the RNA foci seen in those who carry SMAD12 expansions. The manuscript by Florian et al would have benefitted from, at least, an RNA FISH study to determine if the expansions in MARCH6 have the same aggregates as seen in those with SMAD12 expansions.

Whenever there are expansion mutations, one wonders about genotype:phenotype correlation. With 4 families, this may be difficult, but perhaps, given the sample size considerations, determining if the cases with epilepsy have greater repeat expansions compared to those without epilepsy may be do-able.

Reviewer #3:

Remarks to the Author:

Florian et al. builds on the recent findings of intronic repeat expansions in familial adult myoclonic epilepsy loci. This is an excellent paper and when combined with the linked paper helps explain an important cause of FAME, albeit that there are many fewer families reported here and that the essential mechanism was reported over a year ago.

Florian et al. used a variety of repeat expansion analyses of WGS data to review a large European FAME family with linkage to chr5 and finding a segregating polymorphic repeat expansion in intron 1 of MARCH6 similar to those seen in FAME1 and SCA37. Using repeat-primed PCR they then demonstrated that the same expansion segregated well in a previously reported Dutch family and 2 of 11 other European FAME families without linkage studies, with the exception of 3 individuals (two of whom were re-examined and felt to be mildly affected). They quantified the expansion sizes demonstrating that there appears to be somatic mosaicism with micro-rearrangements in lymphocytes, particularly in those individuals with the largest expansions.

Ideally, I would like the descriptive statistics of genotype vs affectation summarised overall and for each family in a table or the text.

Abstract

UK or US spellings? Characterized or characterised? Generalized or generalised? etc

Main text

First line is passive – rephrase with FAME is an.. then follow with ‘first described clause’

We have to trust the authors that they have identified a forme fruste with regards to tremor in 1-IV-8 and 2-V-8. Being as they are critical here it would have been advantageous to see supporting evidence of their affected status. The subclause ‘although’ – does not make sense here – the variably severity supports your assortment that these are clinically affected individuals.

1-IV-8 is interesting – with a number of p alleles in the ‘epilepsy range’ (50+) whereas the CT cases had numbers nearer 30. Can you explain this? Is he one of the ‘paucisymptomatic’ individuals? Do not be tempted to misclassify him, if he really could be normal - state it. This may be important.

Really interesting point about somatic mosaicism and variability of reads by tissues because of the clinical variability. Could you have looked in other tissues?

I like the argument about the pathogenicity leading from transcription of the expansion; how confident can you be that all of these repeat expansion disorders have the same pathogenicity?

I don't fully understand "Interestingly however, similar TTTTA/TTTCA expansions in an intron of 196 DAB1, a gene highly expressed in the cerebellum, have been associated with spinocerebellar ataxia 37 (SCA37)²⁴ 197, demonstrating that the nature of the expansions and their expression in the 198 brain are insufficient to entirely determine the phenotype" – what phenotype? Could not the cerebellum limited expansions cause ataxia and the whole brain pathology cause seizures?

Where does this leave the CTNND2 result which is now neither necessary nor sufficient? Is the argument that it plays a role here – or will the paper be withdrawn?

Are there any clinical ways to differentiate between the FAME1/2 etc? Why is the cause different in Asia? Founder mutation? Could your families be cryptically linked in this way?

Stick with FAME3 – using FAME3/FCMTE3 is not helpful here

Supplementary

442, full stop missing after analyses

Table 1

Do you mean gender (the way they identify in life) or sex (biologically determined)?

Response to comments on manuscript NCOMMS-19-08405-T.

Response to Editor's comments:

We provide with the revised manuscript:

- (1) the updated reporting summary,
 - (2) the source data file containing raw data for each figure included in the article,
 - (3) the data availability statement, including the ClinVar accession entry corresponding to the expansion (page 23 of the main manuscript),
 - (4) clinical data for all four families are available in Supplementary Table 2; NB: detailed clinical data of Families 1 and 3 had previously been published (Magnin et al. Rev Neurol (Paris) 165, 812-820 (2009); Depienne et al. Neurology 74, 2000-2003 (2010); van Rootselaar et al. J Neurol 249, 829-834 (2002)).
- We provide updated paragraphs in the manuscript highlighting clinical findings for specific individuals (see 'Phenotypic variability and genotype-phenotype correlations' paragraphs Pages 9- 11). A short summary of the clinical findings is also available for individuals for whom expansions have been characterized in Table 1 (Page 32).

Response to Reviewers' comments:

Reviewer #1:

1. Multiple studies finding same repeat expansion in different introns leading to same disease strongly indicates TTTTA/TTTCA repeat RNA toxicity in FAME/BAFME. In fact, earlier published studies for the same repeat showed RNA foci/aggregate formation. Although extensive mechanistic studies are beyond the scope of this manuscript, the lack of any data supporting expression of the repeat make these data difficult to interpret. A number of steps could be taken to address this:

- a. Is there any evidence that the TTTCA repeat-containing intron is transcribed or retained in patient cells at increased rates compared to control cells?**
- b. Does the expansion affect mature mRNA and MARCH6 protein expression? These can be easily determined by RT-PCR and western blotting (using commercially available antibodies).**
- c. Perhaps beyond the scope of this manuscript, Does this TTTCA RNA form nuclear foci in patient cells or tissue samples?**

We performed additional experiments to address the comments of Reviewers #1 and #2. We now provide a detailed quantification of the *MARCH6* transcript in both blood cells and fibroblasts of expansion carriers (blood, N=12; fibroblast, N=4) versus non-carriers individuals (blood, N=10; fibroblast, N=4). The results confirm that the expansion has no detectable impact on global *MARCH6* expression in these tissues. In agreement, we also did not observe a difference for MARCH6 protein expression in fibroblasts between control and carrier individuals as evaluated by Western blotting.

Furthermore, we show that the fraction of *MARCH6* mRNA retaining intron 1 is similarly low in expansion carriers and non-carriers, and that there is no detectable increase of intron 1 retention in expansion carriers. These novel findings confirm observations made in lymphoblasts by RNAseq, and have been added into the main text of the revised manuscript (page 11 and 12). The result rules out a massive accumulation of pre-mRNA or unspliced mRNA retaining intron 1 in patients as a consequence of the expansion, which would be expected if RNA foci were present.

Moreover, we think that the current evidence for RNA foci associated with TTTTA/TTTCA expansion is not very strong. RNA foci associated with TTTTA/TTTCA expansions have been reported by Ishiura et al.

(Nat Genet 50, 581-590 (2018)) in post-mortem brains of two individuals with *SAMD12* expansions, including one with biallelic expansions. However, the number of aggregates present in the images (see Supplementary Fig. 9 in the paper) is very low (in fact, restricted to one focus in the heterozygous brain and two in the homozygous brain), which contrasts with what is usually observed with expansion disorders that involve RNA foci, such as myotonic dystrophy, where many, larger, aggregates can be observed (see for example Urbanek et al. Methods 98:115-123 (2016)). We would also like to point out that control treatments with RNase and DNase were not provided to prove that the detected dots really correspond to RNA, and not DNA.

Seixas et al. (Am J Hum Genet 101, 87-103 (2017)) have also reported RNA foci, as a result of TTTTA/TTTCA expansions in *SCA37*, but these results were obtained in the context of overexpression of repeated sequences from a plasmid, which is known to result to non-physiological RNA aggregate artefacts, not representing *in vivo* pathophysiological mechanisms.

So overall, given that RNA FISH is a tricky technique and that we do not have appropriate positive controls for the experiment, we feel that the question raised by the Reviewer needs extensive studies requiring, ideally, human post-mortem brain samples, and therefore should be considered beyond the scope of this study, as anticipated by the Reviewer.

2. Three individuals harboring the repeat expansion reported asymptomatic, two of which later showed mild symptoms. Is there any other factor such as age (25 yrs) or repeat size that can explain this individual (1-V-6)?

We now provide a statement discussing clinical reassessment of formerly asymptomatic individuals (2-V-8, 1-IV-8 and 1-V-6) (pages 10 and 11).

When re-examining the clinical data of individual 1-IV-8 (53 years), we observed that he actually had a single epileptic seizure at age 46 years (he was in fact treated with VPA) and has clear evidence of asymmetric tremor. This led us to reconsider the diagnosis of this individual, who we feel should be regarded as definitely affected. Even when the phenotype clearly is less severe than that of other affected members in this family. The size of the expansion, assessed from blood cells, was similar to that of more severely affected relatives.

The third asymptomatic individual (1-V-6) was unavailable for clinical re-examination. Individual 2-V-8 is still young (30 years) but he does show mild signs of tremor (see Supplementary Fig. 12).

Several non-exhaustive explanations could account for carriers to be less severely affected: (1) the TTTC A part of the expansion could be reduced in brain tissue compared to other tissues, or (2) additional factors than the expansion itself may modulate the severity and penetrance of the disorder. We have now added a sentence in the revised manuscript to address this point (page 11).

Minor points:

3. How many individual samples are used for Oxford Nanopore sequencing? It appears six individuals from 2 families are used, but in the figure legend (Figure 2D) it says five. Also, first individual in the same figure is labeled as 1-III-6_1, which should be 1-IV-6_1.

We thank the Reviewer for noticing the errors in Fig. 2d (now Fig. 3c) and the legend, which have been corrected. A total of six individuals have been analyzed by Oxford Nanopore sequencing but results of only five are displayed in the figure because no read covering the entire expansion was sequenced in the sixth individual. This has now been explained in the figure legend.

4. What is the correlation (r² value) between mean size of the repeat and age of onset of seizure (Figure 3D)? What is the 95% confidence intervals for this correlation? Which individual samples are used to calculate these correlations? These factors should be delineated in Figure 3 legends.

We have added the R² Pearson coefficients and 95% confidence intervals for all correlations shown in Fig. 3d (now Fig. 5c and 5d). Individuals taken into account have been added to the figure legend.

5. Family 2 proband (2-IV-9) sample was used to test mosaicism by molecular combing (Figure 4A), but it is unclear why 2-IV-9 sample was not included in following RP-PCR test (Figure 4C).

RP-PCR assay results for all affected individuals of Families 1 to 4 (including individual 2-IV-9) are displayed in Supplementary Fig. 3a and 3b. The purpose of previous Fig. 4a was to show RP-PCR profiles for individuals where the same assay was positive with different primers to provide additional support for somatic mosaicism. The RP-PCR of individual 2-IV-9 was not displayed in this figure because the assay was not sensitive enough to detect the different expansion configurations detected with molecular combing in this individual. To avoid confusion, we moved the panel to the Supplementary data (now Supplementary Fig. 10).

6. Figure 4E needs more detailed description. It seems observed micro-arrangements in individual 2-IV-9 are noted as 'M' (Stained in Pink color?), but there is no description. Also, in the figure legend 4E appears as 4F and 4F appears as 4G.

We now provide an explanation of M (magenta), which corresponds to the overlay of red (TTTCA) and blue (5' flanking region), suggesting that there is an overlap of probes mapping to these regions in the rearranged allele. We completed and corrected the figure legend (now Fig. 4).

Reviewer #2:

I really have only one comment and that is the lack of experimental evidence for how this expansion causes the disorder FAME3. The authors comment on the RNA foci seen in those who carry SMAD12 expansions. The manuscript by Florian et al would have benefitted from, at least, an RNA FISH study to determine if the expansions in MARCH6 have the same aggregates as seen in those with SMAD12 expansions.

We kindly refer the Reviewer for a detailed answer to our response to point 1 of Reviewer #1. In the response we discuss additional experiments and why we feel that the evidence of reported RNA foci in *SAMD12* is not that strong. Also we provide evidence that we do not see evidence of RNA accumulation with the repeat in cells of patients so would not a priori expect to find RNA foci. Whereas RNA-FISH seems straightforward we consider it quite challenging for the rebuttal and perhaps not that relevant given that we do not observe RNA accumulation in the first place.

Whenever there are expansion mutations, one wonders about genotype:phenotype correlation. With 4 families, this may be difficult, but perhaps, given the sample size considerations, determining if the cases with epilepsy have greater repeat expansions compared to those without epilepsy may be doable.

A statement discussing genotype-phenotype correlations has been added to the revised manuscript (pages 10 and 11). To answer the specific question of the Reviewer, we do see an earlier age at seizure onset associated with larger expansions.

Reviewer #3:

Ideally, I would like the descriptive statistics of genotype vs affectation summarised overall and for each family in a table or the text.

Clinical and genetic data are summarized for all individuals analyzed by Oxford Nanopore sequencing and/or molecular combing in Table 1. We have now also included a statement discussing genotype-phenotype correlations (pages 10 and 11).

Abstract

UK or US spellings? Characterized or characterised? Generalized or generalised? Etc

We have now homogenized the text to US spelling.

Main text

First line is passive – rephrase with FAME is an.. then follow with ‘first described clause’ .

The corresponding sentence was modified.

We have to trust the authors that they have identified a forme fruste with regards to tremor in 1-IV-8 and 2-V-8. Being as they are critical here it would have been advantageous to see supporting evidence of their affected status. The subclause ‘although’ – does not make sense here – the variability supports your assortment that these are clinically affected individuals.

1-IV-8 is interesting – with a number of p alleles in the ‘epilepsy range’ (50+) whereas the CT cases had numbers nearer 30. Can you explain this? Is he one of the ‘paucisymptomatic’ individuals? Do not be tempted to misclassify him, if he really could be normal - state it. This may be important. We kindly refer the Reviewer for a detailed answer to our response to point 2 of Reviewer #1. We now have new evidence that individual 1-IV-8 should be considered definitely affected (he had a seizure at age 46 years, shows clear signs of tremor, and his symptoms are treated with VPA).

Really interesting point about somatic mosaicism and variability of reads by tissues because of the clinical variability. Could you have looked in other tissues?

We have extended the molecular combing analysis to fibroblasts of four expansion carriers from Family 1. The results obtained are comparable to what has been observed previously in the blood of these individuals (see revised Table 1 and Fig. 5b). The data show that the somatic mosaicism is found with both investigated tissues (blood cells and fibroblasts) although none of these four individuals has very large expansions.

I like the argument about the pathogenicity leading from transcription of the expansion; how confident can you be that all of these repeat expansion disorders have the same pathogenicity?

Although one has to be careful to extrapolate findings to other repeat expansion disorders, so we agree with the remark of the Reviewer, we consider it quite striking that a similar expansion in genes with apparently unrelated function result in almost identical phenotypes. Hence, we feel we should make the argument that we favor the explanation that the expansion itself is the cause of disease and not so much its gene location. The statement is also relevant as it may guide efforts to develop a treatment for these diseases. However, as we tried to convey, it is only an (attractive) hypothesis at the moment as the precise mechanisms of what cause disease need to be confirmed in future functional studies.

I don’t fully understand “Interestingly however, similar TTTTA/TTTCA expansions in an intron of 196 DAB1, a gene highly expressed in the cerebellum, have been associated with spinocerebellar ataxia 37

(SCA37)24 197 , demonstrating that the nature of the expansions and their expression in the 198 brain are insufficient to entirely determine the phenotype” – what phenotype? Could not the cerebellum limited expansions cause ataxia and the whole brain pathology cause seizures?

We understand the confusion the sentence may have caused and modified it (page 13). Although speculation at this moment, other genes associated with FAME -including *MARCH6*- are also highly expressed in cerebellum, which makes us think that the gene expression profile is not the sole explanation of the phenotypic discrepancy between FAME and SCA37.

Where does this leave the CTNND2 result which is now neither necessary nor sufficient? Is the argument that it plays a role here – or will the paper be withdrawn?

The identification of an expansion in Family 3 has indeed led to a reclassification of the *CTNND2* variant as likely benign. We have added a sentence to make this clear (page 7).

We would like to point out that the results presented in the van Rootselaar et al. (Neurology 89, 2341-2350 (2017)) paper are still accurate (i.e. the *CTNND2* variant reported truly exists and the functional study clearly shows an impact of this variant on neuronal morphology in vitro). The conclusion was therefore the logical interpretation of the data at the time, also given that expansions could not be detected by the methods used in this paper. In addition, claims that specific genes were involved in other FAME/FCMTE loci had been made by several other groups with the same methodology and limitations. Overall, we see no ground to retract the paper and think that additional studies are required to understand the real impact of this variant in the Dutch family.

Are there any clinical ways to differentiate between the FAME1/2 etc?

There is little difference in phenotype that can be attributed to the locus itself, i.e. FAME1, -2 or -3. The only thing that may point to a clinical difference is that seizures in FAME3 seem to occur, on average, a bit earlier in disease development, i.e. seizures are present at disease onset in many patients while in other FAME forms, tremor usually precedes seizures by several years.

Why is the cause different in Asia? Founder mutation? Could your families be cryptically linked in this way?

There is a known founder effect for the FAME1 chromosome 8 locus explaining why this locus is more frequent in the Asian population. We have performed an analysis of the haplotype shared by the two French families. The results support the existence of an ancient haplotype underlying FAME3 expansions as now mentioned in a new paragraph (page 7).

Stick with FAME3 – using FAME3/FCMTE3 is not helpful here

We have removed FCMTE3 from the corresponding sentence.

Supplementary

442, full stop missing after analyses

We checked that full stops terminate every sentence in the revised Supplementary Material.

Table 1

Do you mean gender (the way they identify in life) or sex (biologically determined)?

We replaced “gender” by “sex”.

Reviewers' Comments:

Reviewer #1:

Remarks to the Author:

Overall the manuscript is acceptable.

I would note, however, that I would encourage the authors to at least publish an erratum on their manuscript in *Neurology* (van Rootselaar et al.) to avoid confusion going forward. Not everyone will deep-dive into this paper enough to see their footnote on the subject and it will be important for others who identify this non-repeat variant in the future.

Reviewer #2:

Remarks to the Author:

No further comments. The authors have addressed my comments regarding functional experiments.

Reviewer #3:

Remarks to the Author:

The authors have produced a comprehensive and helpful response to all three reviews. Their explanations are clear and improve my understanding of their work. This is a high standard report and one that advances the field significantly.

Response to Reviewers' comments on manuscript NCOMMS-19-08405A.

Reviewer #1:

Overall the manuscript is acceptable.

I would note, however, that I would encourage the authors to at least publish an erratum on their manuscript in *Neurology* (van Rootselaar et al.) to avoid confusion going forward. Not everyone will deep-dive into this paper enough to see their footnote on the subject and it will be important for others who identify this non-repeat variant in the future.

We thank the reviewer for the suggestion to publish an erratum. This possibility will be considered with all coauthors of the *Neurology* paper, some of whom are not involved in the present study, and eventually discussed with the editors of *Neurology*.

We think that this action is unrelated to the final acceptance of the present paper, which already provides the necessary information about the classification of the *CTNND2* variant.

We would like to emphasize that publication of errata should also be suggested and coordinated for other papers wrongly reporting possibly pathogenic variants on chromosomes 2 (De Fusco et al, *Ann Neurol* 2014) and 8 (Lin et al, *Neurosci Lett.* 2018).

Reviewer #2:

No further comments. The authors have addressed my comments regarding functional experiments.

Reviewer #3:

The authors have produced a comprehensive and helpful response to all three reviews. Their explanations are clear and improve my understanding of their work. This is a high standard report and one that advances the field significantly.

We thank reviewers #2 and #3 for their positive comments.